# Occupational inequalities in the prevalence of COVID-19: A longitudinal observational study of England, August 2020 to January 2021

**Mark A. Green**[1]*, **Malcolm G. Semple**[2]

**1** Department of Geography & Planning, University of Liverpool, Liverpool, United Kingdom, **2** NIHR Health Protection Research Unit in Emerging and Zoonotic Infections, Institute of Infection Veterinary and Ecological Sciences, University of Liverpool, Liverpool, United Kingdom

\* mark.green@liverpool.ac.uk

## Abstract

The COVID-19 pandemic has reinforced, amplified and created new health inequalities. Examining how COVID-19 prevalence varies by measures of work and occupation may help to understand these inequalities. The aim of the study is to evaluate how occupational inequalities in the prevalence of COVID-19 varies across England and their possible explanatory factors. We used data for 363,651 individuals (2,178,835 observations) aged 18 years and over between 1st May 2020 and 31st January 2021 from the Office for National Statistics Covid Infection Survey, a representative longitudinal survey of individuals in England. We focus on two measures of work; employment status for all adults, and work sector of individuals currently working. Multi-level binomial regression models were used to estimate the likelihood of testing positive of COVID-19, adjusting for known explanatory covariates. 0.9% of participants tested positive for COVID-19 over the study period. COVID-19 prevalence was higher among adults who were students or furloughed (i.e., temporarily not working). Among adults currently working, COVID-19 prevalence was highest in adults employed in the hospitality sector, with higher prevalence for individuals employed in transport, social care, retail, health care and educational sectors. Inequalities by work were not consistent over time. We find an unequal distribution of infections relating to COVID-19 by work and employment status. While our findings demonstrate the need for greater workplace interventions to protect employees tailored to their specific work sector needs, focusing on employment alone ignores the importance of SARS-CoV-2 transmission outside of employed work (i.e., furloughed and student populations).

## Introduction

The social, health and economic impacts resulting from the spread of Severe Acute Respiratory Syndrome Coronavirus-2 (SARS-CoV-2), and restrictions aimed at managing its spread, have been unprecedented in scale and scope. In England, as in many countries, the impacts of Coronavirus disease 2019 (COVID-19) resulting from SARS-CoV-2 have been unevenly felt across populations. Hospitalisation and mortality outcomes related to COVID-19 have been higher

**Data Availability Statement:** The ONS's COVID-19 Infection Survey (CIS) is an individual level survey that collects population level data about people who live in the UK to estimate the prevalence of

COVID-19. They collect a range of data including demographic and social data, as well as test everyone for COVID-19. These data contain sensitive and personal information about individuals (including information about date of birth and health information). Such data cannot be openly shared to ensure privacy and meet statistical disclosure laws (i.e., researchers cannot legally distribute). Access requires approval by the ONS who act as the data owner and do not allow researchers to hold or distribute their data. Data are free to access for accredited researchers within the ONS's Secure Research Service (SRS). The SRS is a trusted researcher environment which provides free access to the ONS's sensitive data (it acts as a secure data repository). More information about the process, including how to apply to access the data, can be found here: https://www.ons.gov.uk/aboutus/whatwedo/statistics/requestingstatistics/secureresearchservice All analytical code used to process the data and replicate the results in the paper can be found here: https://github.com/markagreen/occupational_inequalities_CIS Metadata: https://ons.metadata.works/browser/dataset?id=293.

**Funding:** This work was supported by the Economic and Social Research Council [grant number ES/L011840/1]. The funders had no role in study design, data collection and analysis, decision to publish, or preparation of the manuscript.

**Competing interests:** The authors have declared that no competing interests exist.

in older populations, males, Black and Asian ethnic groups, and deprived communities [1–6]. Understanding and tackling the social inequalities arising from and amplified by COVID-19 remains a core UK Government priority.

As our personal and social lives have had to adapt to the COVID-19 pandemic, so too has our economic, work and employment circumstances to minimise the transmission of SARS-CoV-2. Many employment roles were 'furloughed' (i.e., temporary unemployment), with salaries being covered by the UK Government if an employer did not terminate jobs. Some occupation roles adapted so that individuals could work from home, whereas others were able to introduce protective social distancing measures. However, not all occupations were able adapt to either of these strategies resulting in different population-level exposures to SARS-CoV-2.

Emerging evidence has demonstrated that 'essential' occupations who work directly with patients (e.g., health or social care workers), groups unable to work from home (e.g., transport or manufacturing occupations), or occupations with 'front facing' roles where individuals are routinely exposed to others (e.g., supermarket workers or teachers) were at greater risk of severe COVID-19 outcomes including mortality [7–14]. Most of this existing research has focused on severe outcomes or a narrow range of occupations/work sectors, meaning that we have less evidence of how SARS-CoV-2 infections vary across occupations or work settings. Preventing infections and exposure to SARS-CoV-2 will help to reduce severe COVID-19 outcomes. Understanding which occupations had higher or lower infections is imperative for designing preventative work-place interventions for managing COVID-19 and preparing for future pandemics.

Occupational status intersects with age, sex, ethnicity and deprivation. For example, individuals from deprived neighbourhoods or Black and Asian ethnic groups are more likely to be employed in occupations that were unable to work from home or in 'essential' front facing roles [3, 15]; the same groups who have seen higher hospitalisations and mortality related to COVID-19. Occupation type is therefore a fundamental driver of exposure to SARS-CoV-2 in employed populations, meaning that occupation may partly explain or amplify health inequalities relating to COVID-19 [5, 16]. As such, the ability of work to adapt or change within government restrictions was experienced unevenly across the population and may partly explain the pathways through which social inequalities in COVID-19 have materialised. Identifying which work sectors are at highest risk can help us to design social distancing and preventative measures beyond vaccination that could potentially narrow health inequalities [17].

The aim of this study is to evaluate how occupational inequalities in the prevalence of COVID-19 vary across England and their possible explanatory factors. We address several limitations in the literature investigating this issue. First, there is a paucity of evidence on the extent of risks of COVID-19 by granular occupational groups or work sectors. Second, evidence derived from 'testing' data are biased due to self-selection (i.e., focusing on individuals with symptoms or who are engaged in testing). We tackle this through using a novel survey where participants were all tested irrespective of symptoms. Third, our large survey helps accommodate issues relating to the rarity of events, allowing for detailed investigations into occupational inequalities in COVID-19. Finally, we consider how trends in occupational inequalities changed throughout our study period to examine if certain time periods produced differences in infection risk across occupational groups.

## Materials and methods

### Data

The Office for National Statistics (ONS) Covid Infection Survey (CIS) was used as our data source. The CIS is a representative random sample survey of the population in England used

to monitor trends in COVID-19 [18]. While primarily used as a surveillance tool for COVID-19, CIS was also designed so that the data could be re-used by researchers as a secondary data resource for understanding population health issues relating to the pandemic. Secondary data are useful resources since they allow for efficient collection of large and complex data that saves time and costs over primary data collection [19]. This is valuable for our study since (i) the rarity of our outcome (0.9% in this study) means we require a large sample size to find robust associations, and (ii) the CIS was designed to collect a national-level representative sample that minimises selection bias meaning our findings can be generalisable.

In the CIS, individuals are invited take a SARS-CoV-2 test irrespective of whether they have symptoms or not, allowing an estimate of overall COVID-19 prevalence. This helps to minimise issues in secondary data based on self-reported testing records due to selection bias in who tests and who registers tests [20]. Individuals also complete a survey about a range of demographic, social and health questions that contextualise their circumstances. The survey and SARS-CoV-2 tests were completed together at each data collection time point. There were 2,772,698 observations between 1st May 2020 and 31st January 2021 available for analysis.

Data from August onwards (n = 2,518,142) were selected to assess trends during the second wave of infections in England. While CIS started in May, the low infection levels and limited data collection between May and August meant we removed these data. Although August also has low levels of infections, it was included to capture any signals preceding the start of the second wave occurring in September. Only observations for adults aged 18 years and over were selected for the analysis (n = 2,178,835).

While the CIS is a repeated cross-sectional survey, individuals were encouraged to take part repeatedly over time. Participants were asked to enrol for follow-up waves, initially weekly over the first month then monthly up to 13 times [18]. Our study leverages this longitudinal design. Attrition from the survey was low, reported as 0.62% in December 2020, supported by monthly data refreshes of new participants in response to the attrition and ensure the representativeness of the sample [18]. The analytical sample contained 363,651 individuals. 61% of individuals had at least one record per month (mean number of records over the study period was 6, with a standard deviation of 2.2). 5% of individuals had only one occurrence in the data. We utilise each observation as the primary unit of analysis, nested within individuals.

The outcome variable for our analysis was whether an individual had a positive SARS-CoV-2 test or not (binary). Tests were nose and throat swabs self-administered by participants and posted for analysis at a hub laboratory. Swabs were tested for SARS-CoV-2 using reverse transcription polymerase chain reaction (PCR) tests [18]. Tests recorded as 'void' or 'insufficient' were excluded from the analysis.

We selected two measures of occupation and work as our primary measures of interest. First, *employment status* was chosen to represent an individual's primary employment circumstances as an aggregate measure of work-related risk. Categories were employed, self-employed, furloughed (i.e., individuals temporarily not working), student, or not working (e.g., retired, economically inactive, unemployed). Second, focusing on just individuals who are currently working either employed or self-employed, we also consider *work sector*. 15 categories of *occupational sectors* (e.g. teaching and education, health care, retail sector) were used to assess differences in COVID-19 risk by type of work. Low sample sizes for specific occupations meant they were less suitable for the analysis. Descriptions of each work sector can be found in Table A in S1 File.

Additional explanatory variables were selected based on key factors that may help to explain occupational differences in COVID-19 included:

- *Age*–included to assess differences in risk by age, due to evidence that younger population groups were more likely to have a positive SARS-CoV-2 test and older age groups more likely to have experienced severe harms relating to COVID-19 [2, 4]. Age is used both in its raw value, as well as squared to account for possible non-linear effects.

- *Sex*–included to assess the differences in factors affecting males and females differently [2, 4].

- *Ethnicity*—selected due to inequalities in social and health impacts of COVID-19 disproportionally affecting non White British populations [1–3, 6]. Ethnic groups were kept as specific groups where possible, although some groups were combined together to ensure sufficient sample sizes and avoid data disclosure issues.

- *Number of people within a household*–chosen as a greater number of people may increase opportunities for the spread of SARS-CoV-2 [21].

- *Whether an individual had travelled abroad recently or not*–included due to the possible higher risk from individuals travelling to countries with higher COVID-19 prevalence or greater social mixing [22].

- *Work location*–for analyses using work sector only, we account for whether individuals were working at home, outside the home or a mixture of both. Individuals who are working outside of the home may have higher risk as they may be mixing with other individuals or have greater exposure to SARS-CoV-2 [13].

- *Month*–we also adjust for month of the year to account for the differential risk of COVID-19 that varied over time, although we do not report these results.

## Statistical analyses

Descriptive summary statistics and visualisations are used to describe aggregate patterns in the data. We first describe demographic patterns in our outcome variable to help contextualise our statistical analyses. Multi-level binomial regression models were used to analyse the risk of COVID-19. Two models were used; one for employment status and one for occupational sector (our key exposure variables). Multivariable models were fully adjusted using a series of fixed effect variable representing our explanatory individual-level covariates that may explain differences in COVID-19 risk. Numeric values were z-score standardised (age and household size). Two random effects are included: (i) participant ID (varying intercept) to account for repeat observations within the survey over time, and (ii) geographical area of residence (varying intercept). While we do not report results by geographical location here, it was included to account for the spatial heterogeneity in COVID-19 outcomes observed in England [2]. Regions (n = 116) were created by the ONS and match Local Authority districts, with districts combined to make sure no region has a population of less than 500,000 to preserve data security. We present only the fully adjusted analyses here.

Finally, we also considered how associations changed in each month through introducing a series of interaction effects into the above models (i.e., interactions between month and other fixed effects). This was done to examine if any occupational group experienced differences in infections at certain time periods which might have not been consistent across work sectors (e.g., differences in restrictions or lockdowns affecting who could work from home). As such, it can be difficult to interpret the results. To aid the interpretation, we calculated the predicted probability (presented as percentages to aid interpretation) of each occupational measure

testing positive for COVID-19 for each month by sex (adjusting for other covariates in the model). The models are supplemented through additional analyses stratifying the model by age group (defined as individuals aged less than 40 years, and individuals aged 40 years and over).

## Results

### Demographic inequalities in COVID-19 prevalence

Summary sample characteristics can be viewed in the (S1 File). 0.9% of respondents tested positive for COVID-19 during the study period. Fig 1 presents trends in the estimated prevalence of COVID-19 during the study period. Following low levels of COVID-19 in August, prevalence of COVID-19 began to rise in September onwards peaking in the first week of November. COVID-19 prevalence declines thereafter, following a national lockdown on 5th November 2020, before rising again after the end of the lockdown (2nd December 2020) and

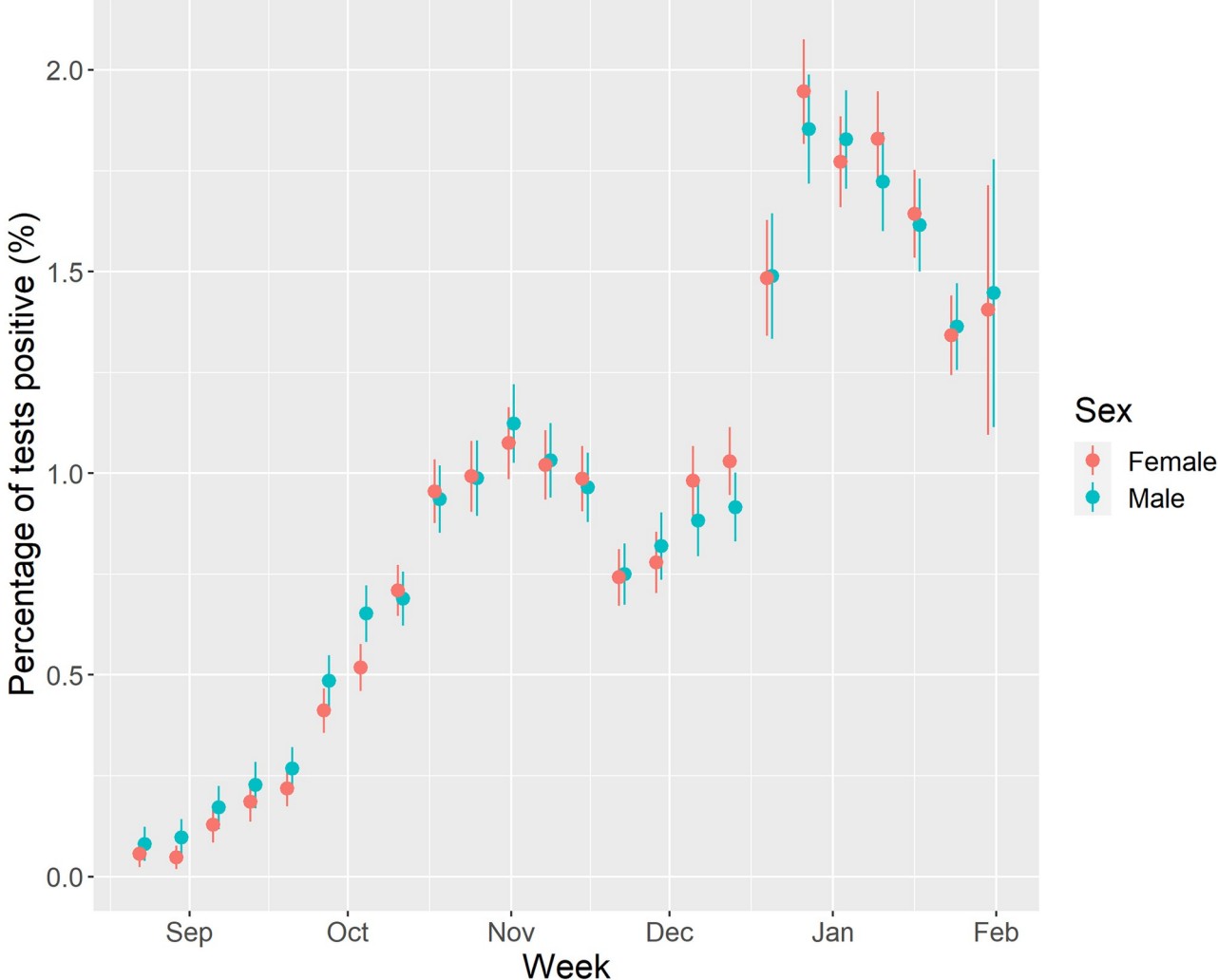

**Fig 1. Percentage of tests that were positive for COVID-19 by week of year and sex (unadjusted).** (Note: figures in first two weeks of August were redacted due to disclosive numbers (i.e., <10 positive tests. Point is estimated percentage, with error bars the 95% confidence intervals).

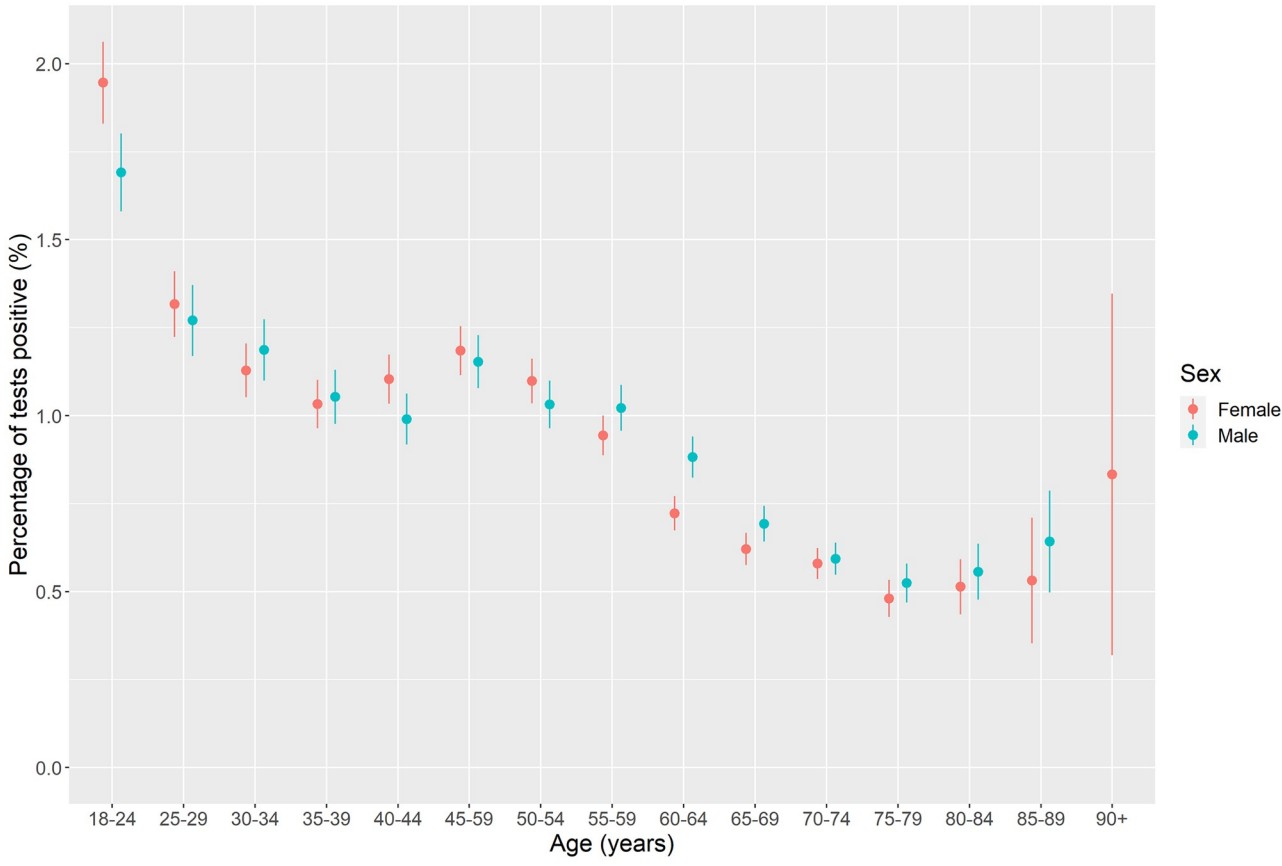

**Fig 2. COVID-19 prevalence by 5-year age band and sex (unadjusted).**

with the emergence of the B1.1.7 (Alpha) variant. Trends then decline following the national lockdown announced on the 6[th] January 2021. There were no noticeable differences in trends between males and females.

Fig 2 examines how COVID-19 prevalence varies by single year of age. Highest prevalence of COVID-19 was among ages 18–24 with prevalence almost twice as high as the national average. Prevalence was higher among these ages for females compared to males. Prevalence of COVID-19 declines with age thereafter with minimal differences by sex.

Fig 3 presents COVID-19 prevalence by ethnic group. Highest prevalence of COVID-19 was found for Pakistani ethnicity (more than two times higher than the prevalence of the White British group), with higher prevalence also among Black African groups. Lower prevalence was observed for Chinese, White British, and Mixed White and Asian groups. There were no significant differences by sex.

### Descriptive inequalities by employment type and work sector

Fig 4 presents COVID-19 prevalence among all adults by employment status. The highest prevalence of COVID-19 was observed for individuals who were students or furloughed (i.e., individuals temporarily not working). Lower prevalence was estimated for individuals who were not working (i.e., retired, unemployed, long-term sick) and self-employed groups. There were no differences by sex. Stratifying analyses by age revealed higher prevalence of COVID-

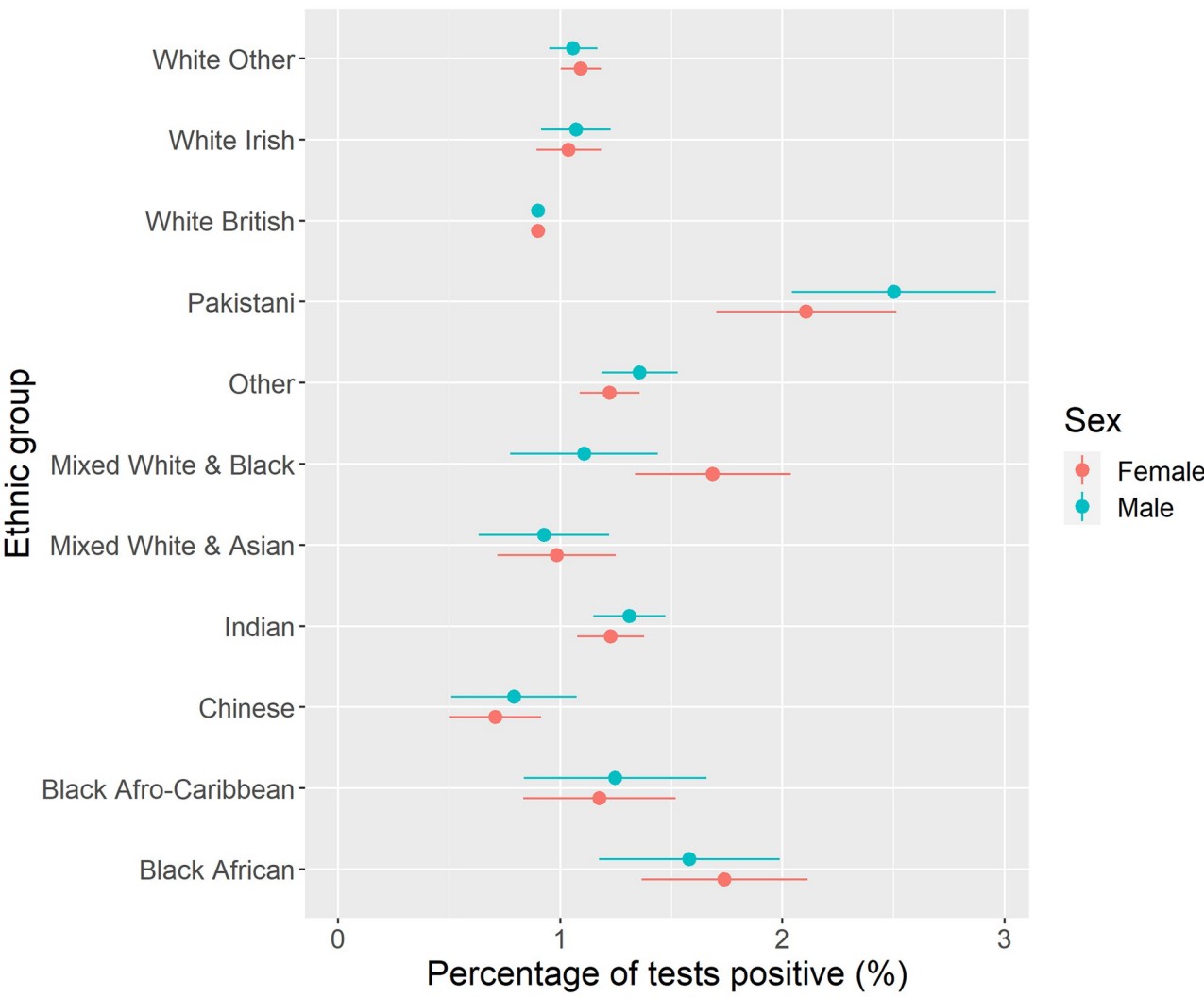

**Fig 3. COVID-19 prevalence by ethnic group and sex (unadjusted).**

19 among younger populations for each employment type, especially younger furloughed males (S1 File).

We next considered inequalities in prevalence of COVID-19 by work sector (Fig 5). The highest prevalence of COVID-19 overall was found for individuals employed in the hospitality sector, with higher prevalence for individuals employed in transport, social care, retail, health care and education. Lowest prevalence was for individuals employed in ICT, with low prevalence in the armed forces and entertainment sectors. Stratifying analyses by age showed higher risks across most work sectors for younger populations (S1 File).

## Regression analyses of occupational risk of COVID-19

First, we examined the likelihood of having a positive SARS-CoV-2 test by employment status (Table 1). Here we describe groups of person-time as the measure is time dependent. In comparison to individuals who were employed, individuals who were furloughed had 81% higher odds (Odds Ratio (OR) = 1.81, 95% Confidence Intervals (CIs) = 1.69–1.93) of a positive

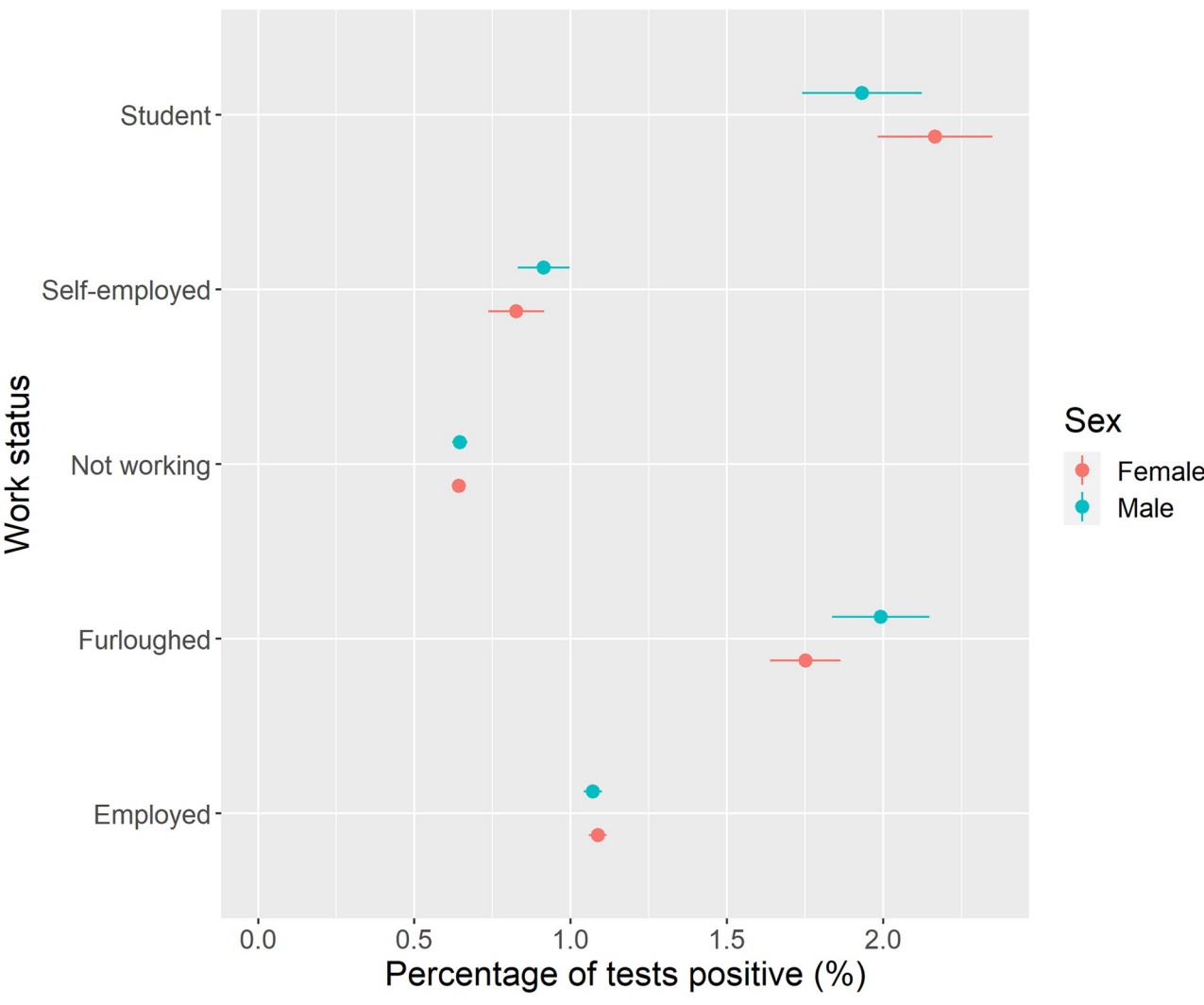

**Fig 4. COVID-19 prevalence by work status and sex (unadjusted).**

SARS-CoV-2 test. Students had 35% (OR = 1.35, 95% CIs = 1.22–1.50) higher odds than employed individuals for a positive SARS-CoV-2 test.

There were distinct demographic inequalities. Age was negatively associated with COVID-19 risk, so that older populations were less likely to have tested positively. No association for sex was detected. Ethnic inequalities were evident, with greater risk of COVID-19 found for Indian (13% higher odds), Pakistani (69%), Black African (36%) and White Irish (20%) populations than compared to White British populations. We observed a greater likelihood of COVID-19 among individuals who had travelled abroad recently. Finally, there was a positive association to number of people in the household, suggesting greater prevalence of COVID-19 among larger households.

Table 2 presents our second model which considers only individuals currently working (i.e., excluding groups who were students, furloughed, or were not working) who were employed or self-employed. Here we describe groups of person-time as the measure is time dependent. In comparison to individuals employed in ICT occupations, individuals employed

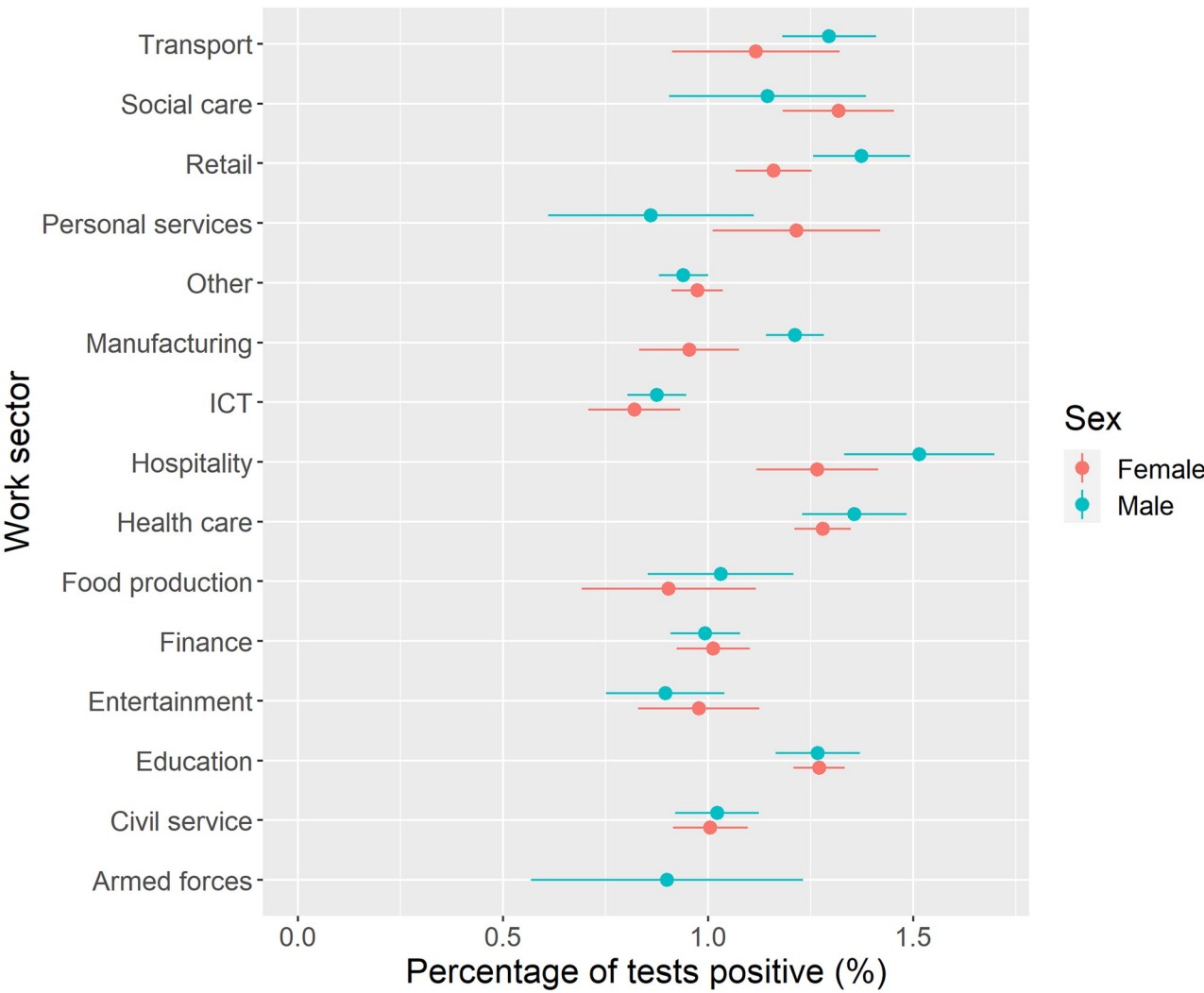

**Fig 5. COVID-19 prevalence by work sector and sex (unadjusted).** Note: estimate for females employed in the armed forces excluded due to counts <10 to preserve ONS data disclosure standards.

in education (27% higher odds), health care (29%), social care (43%), transport (19%), retail (23%), hospitality (30%) and manufacturing (18%) were more likely to have a positive SARS-CoV-2 test. The analysis also accounts for where individuals are working from. Individuals who are unable to work from home had 30% (OR = 1.30, 95% CIs = 1.23–1.38) higher odds of a positive SARS-CoV-2 test than compared to individuals who were working from home.

The model also finds similar associations for age, household size and whether an individual had travelled abroad as reported previously. Fewer associations were detected by ethnic group, although individuals of Pakistani ethnicity had 75% (OR = 1.75, 95% CIs = 1.37–2.24) higher odds to have a positive SARS-CoV-2 test than compared to individuals of White British ethnicity.

### Estimating change over time in COVID-19 prevalence by work status

The final section of our analysis extends the regression analysis presented in the above section to consider how the relationships and associations vary by month.

**Table 1. Model summary for analysing COVID-19 risk by socio-demographic features including occupational status.**

| Variable | Odds Ratio | Lower CI | Upper CI |
|---|---|---|---|
| Male | Reference | | |
| Female | 0.994 | 0.958 | 1.031 |
| Age (z-score) | **0.858** | **0.763** | **0.965** |
| Age-squared (z-score) | 1.003 | 0.884 | 1.138 |
| White British | Reference | | |
| Any other ethnic group | 1.102 | 0.985 | 1.235 |
| Any other white background | 0.987 | 0.903 | 1.080 |
| Chinese | 0.624 | 0.468 | 0.833 |
| Indian | **1.130** | **1.004** | **1.273** |
| Pakistani | **1.687** | **1.389** | **2.048** |
| Black African | **1.359** | **1.086** | **1.700** |
| Black Afro-Caribbean | 1.123 | 0.848 | 1.486 |
| Mixed White & Asian | 0.874 | 0.675 | 1.131 |
| Mixed White & Black | 1.190 | 0.947 | 1.496 |
| White Irish | **1.204** | **1.025** | **1.415** |
| Employed | Reference | | |
| Self-employed | 0.927 | 0.851 | 1.011 |
| Furloughed | **1.808** | **1.692** | **1.933** |
| Not working | 0.834 | 0.790 | 0.880 |
| Student | **1.350** | **1.216** | **1.500** |
| Have travelled abroad recently | **1.132** | **1.072** | **1.196** |
| Household size (z-score) | **1.124** | **1.104** | **1.145** |
| Random effects | Variance | SD | |
| ID (participant) | 3.149 | 1.774 | |
| Geographical area | 0.130 | 0.360 | |

Note: Model also adjusted for time (month). Results placed in bold to emphasise associations where 95% confidence intervals (CIs) do not contain 1.

First, we consider the work status of all adults (Fig 6). Increasing predicted probability of COVID-19 is observed through the period for all groups other than students, who experienced a higher predicted probability in October before declining thereafter and remaining flat over the remaining months of the study period. The highest predicted probability for each month was otherwise predicted for furloughed individuals, with individuals who were not working having the lowest probability in each month. Stratifying the analyses by age group suggests no difference in estimated risk across each work status group albeit with the wide uncertainty in estimates (S1 File). The analysis does reveal the higher than expected prevalence of COVID-19 predicted for both furloughed males under 40 years and females under 40 years not working in January.

Fig 7 presents the next model, analysing the predicted probability of COVID-19 by work sector among adults who were currently working. A similar trend of higher predicted probability is observed over time and is consistent by work sector. Differences in the predicted probability of COVID-19 between work sector largely follow the results presented in Table 2, with the highest predicted probabilities in January observed for individuals working in transport, hospitality, retail health and social care. One noticeable difference to the general trend is the

**Table 2. Model summary for analysing COVID-19 risk by socio-demographic features including occupational group for adults who work.**

| Variable | Odds Ratio | Lower CI | Upper CI |
|---|---|---|---|
| Male | Reference | | |
| Female | 0.958 | 0.909 | 1.009 |
| Age (z-score) | **0.724** | **0.578** | **0.908** |
| Age-squared (z-score) | 1.235 | 0.941 | 1.622 |
| White British | Reference | | |
| Any other ethnic group | 1.076 | 0.932 | 1.242 |
| Any other white background | 0.975 | 0.875 | 1.086 |
| Chinese | 0.718 | 0.510 | 1.010 |
| Indian | 1.111 | 0.959 | 1.287 |
| Pakistani | **1.749** | **1.368** | **2.237** |
| Black-African | 1.276 | 0.970 | 1.678 |
| Black Afro-Caribbean | 1.039 | 0.723 | 1.494 |
| Mixed-White & Asian | 0.894 | 0.650 | 1.229 |
| Mixed-White & Black | 1.124 | 0.834 | 1.515 |
| White-Irish | 1.065 | 0.855 | 1.327 |
| ICT | Reference | | |
| Teaching and education | **1.274** | **1.135** | **1.429** |
| Health care | **1.285** | **1.140** | **1.450** |
| Social care | **1.428** | **1.207** | **1.689** |
| Transport (incl. storage, logistic) | **1.192** | **1.025** | **1.386** |
| Retail sector (incl. wholesale) | **1.230** | **1.081** | **1.401** |
| Hospitality (e.g. hotel, restaurant) | **1.300** | **1.092** | **1.547** |
| Food production, agriculture, farming | 0.936 | 0.743 | 1.179 |
| Personal services (e.g. hairdressers) | 1.035 | 0.803 | 1.334 |
| Financial services incl. insurance | 1.062 | 0.940 | 1.200 |
| Manufacturing or construction | **1.176** | **1.042** | **1.326** |
| Civil service or Local Government | 1.118 | 0.980 | 1.274 |
| Armed forces | 0.998 | 0.636 | 1.566 |
| Arts, Entertainment or Recreation | 1.022 | 0.851 | 1.227 |
| Other occupation sector | 1.037 | 0.928 | 1.159 |
| Work from home | Reference | | |
| Working somewhere else (not your home) | **1.302** | **1.231** | **1.376** |
| Both (from home and somewhere else) | **0.890** | **0.817** | **0.970** |
| Have travelled abroad recently | **1.147** | **1.072** | **1.227** |
| Household size (z-score) | **1.088** | **1.064** | **1.113** |
| Random effects | Variance | SD | |
| ID (participant) | 2.943 | 1.716 | |
| Geographical area | 0.147 | 0.384 | |

Note: Model also adjusted for time (month). Results placed in bold to emphasise associations where 95% confidence intervals (CIs) do not contain 1.

higher probability in September for women employed in personal services. Stratifying by age group (S1 File) suggests this additional risk is concentrated among women less than 40 years (predicted value = 1.23%, 95% CIs = 0.14%–2.32%). Similarly, this group observes a large jump in the general trend in December (predicted value = 1.74%, 95% CIs = 0.61%–2.83%) compared to other sectors (additionally concentrated among younger adults).

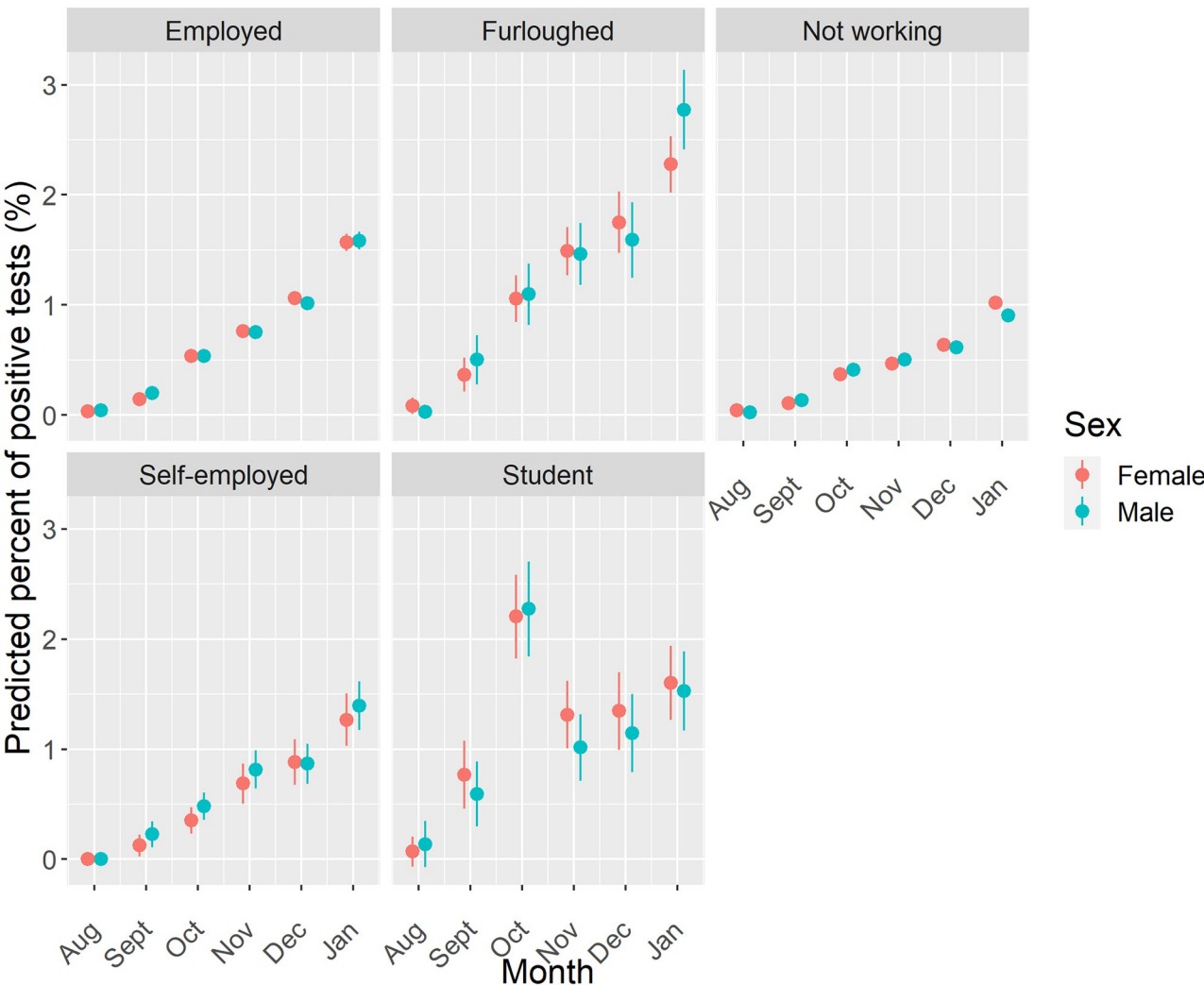

**Fig 6. Predicted probability of testing positive for COVID-19 by work status, sex and month.** Note: Estimates are adjusted for geographical location, age, ethnicity, household size and whether an individual had travelled abroad.

## Discussion

Our study presents one of the most detailed investigations into the extent of occupational inequalities in COVID-19 for England. For all adults, individuals who were furloughed (i.e., temporarily not working) or students had higher prevalence of COVID-19. Focusing on adults currently working, individuals employed in health or social care, retail, personal services, transport, hospitality and teaching had higher likelihood of testing positive for COVID-19. We also find demographic inequalities with COVID-19 prevalence being higher among younger populations and Pakistani, Black African and Indian ethnic groups. Our findings demonstrate the need to tackle the social determinants of COVID-19 to equitably manage the pandemic–with work status and occupational group being one route for interventions aimed at tackling inequalities.

The finding that COVID-19 was higher among furloughed populations may initially feel counter-intuitive, since individuals may have found it easier to socially distance or isolate

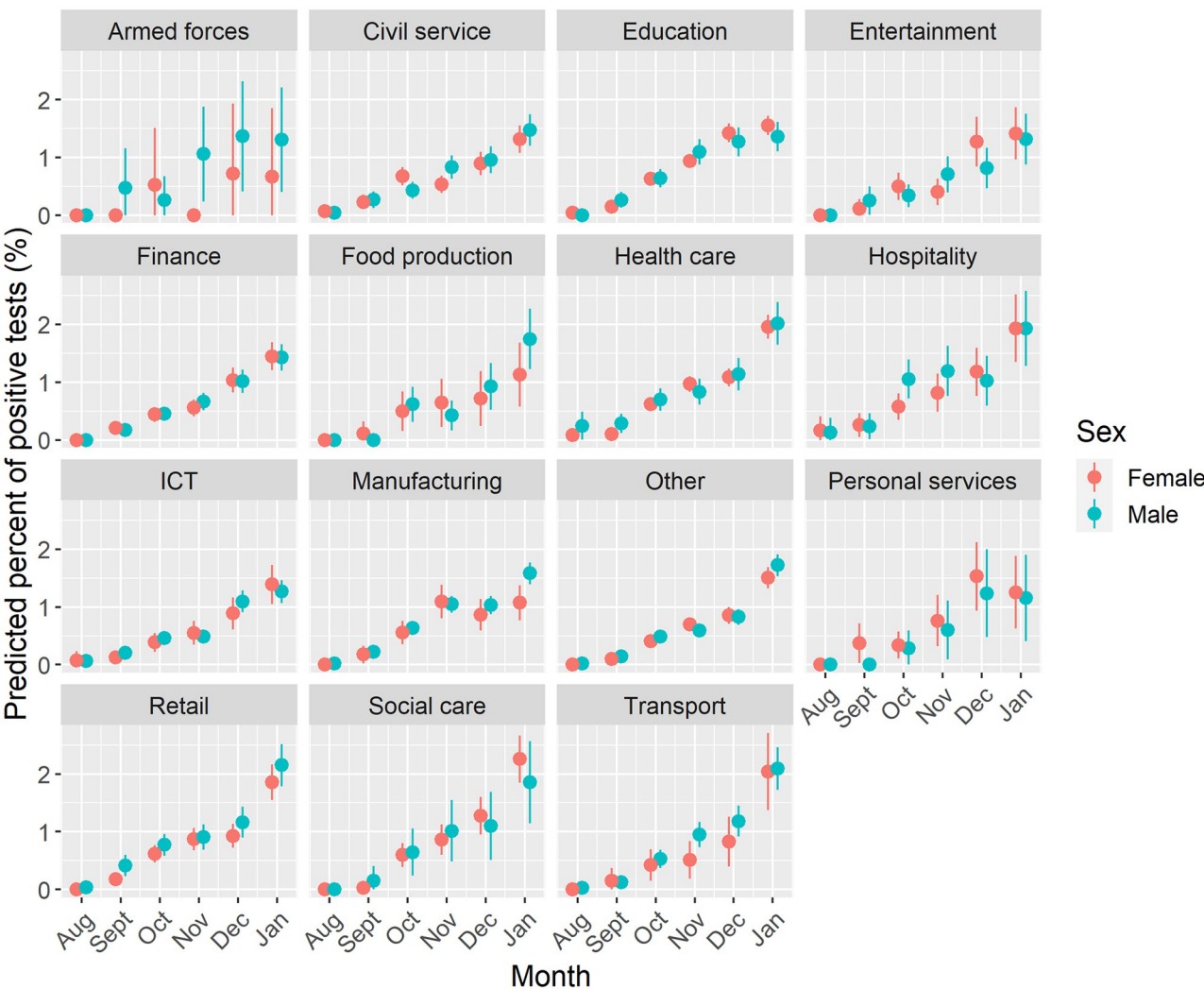

**Fig 7. Predicted probability of testing positive for COVID-19 by work sector, sex and month.** Note: Estimates for geographical location, age, ethnicity, household size, work location and whether an individual had travelled abroad.

compared to those groups employed. While furloughed populations may have fewer work social contacts, evidence suggests that their leisure and social contacts were higher than other groups [23]. Evidence has showed that people who were low-income backgrounds were more likely to have been furloughed [15]. It suggests a 'double jeopardy' effect whereby individuals are not just negatively impacted by being furloughed (e.g., economic hardship from lost labour opportunities or stress from fear of eventual unemployment), but are also more likely to develop COVID-19 that may doubly disadvantage their health and wellbeing. Evidence elsewhere has suggested that people who were furloughed also experienced a small decline in their mental health [24]. Our findings would recommend careful educational messaging where future furlough schemes are introduced to help guide individuals who were furloughed in navigating their new risks, or encouraging employers to find ways to help individuals work from home rather than being furloughed where possible.

A similar explanation for the importance of social mixing can help to explain the high prevalence among students. The large spike in prevalence observed in October coincides with the

start of most University terms where social mixing of individuals from different regions would have occurred. University student migration represents the largest annual internal migration flow in England [25], and managing the process safely will be important to minimising further outbreaks. Improved ventilation and use of marks in indoor classroom settings during periods of high community prevalence may be needed to avoid similar patterns during term time [26]. Such interventions may only be effective if paired with strategies around the social experiences of University students.

Among working individuals, we find that COVID-19 prevalence was not equitably spread across work sectors. Higher prevalence of COVID-19 was not just in patient or care focused professions (e.g., health or social care sectors). We also found that COVID-19 was more common among individuals in work sectors characterised by roles less able to work from home or with greater exposure due to social mixing (e.g., transport, hospitality, retail, personal services or teaching). Our findings follow similar evidence for older adults (50–64 years) on the higher risk of COVID-19 and severe outcomes for key workers [7], patterns in national testing records [9], as well as for occupational inequalities in COVID-19 mortality [8, 10]. Importantly, we add to this literature through tentatively demonstrating how occupational inequalities were not consistent over time, with tentative evidence of 'seeding' of COVID-19 transmission among individuals (particularly young females) employed in personal services (e.g., hairdressers) at the start of the second wave. We also suggest caution in any interpretation that there might be different associations between occupational categories over time. Most measures of work/occupation we analysed closely followed overall population-level trends (i.e., there was no sequencing of transmission between occupational groups). Time-specific interventions targeted at particular occupational groups may therefore be ineffective or unfeasible.

Our findings suggest the need for better workplace interventions across diverse roles that can help contain COVID-19 transmission, whilst allowing individuals and employers to continue their social and economic activities [7]. Occupational roles will need to further adapt to protect their employees from COVID-19. Minimising social contacts or mixing within occupational roles through sufficient preventative measures may be valuable. One study suggested that limiting the number of social contacts at work was the most important strategy for lowering the 'R' number if society keeps schools open [17]. Repeat testing of employees may help to manage outbreaks, however testing behaviours can also widen inequalities [20]. Support for lost earnings if individuals have to self-isolate will be key, especially as some of the work sectors identified here with higher prevalence (e.g., retail or hospitality) are characterised by low wages [16]. The UK Government should also consider targeting particular work sectors considered at 'higher risk' of infections and introducing interventions in those areas when required. However, our findings of high prevalence of COVID-19 for furloughed and student populations demonstrates the need for broader strategies than just occupation-related interventions to help manage COVID-19 and tackle the drivers of health inequalities.

Our analyses also demonstrate wide ethnic inequalities in COVID-19 prevalence. Likelihood of having had a positive COVID-19 test was higher among Pakistani, Black African, and Indian groups than compared the majority White British population. Our results follow evidence from other studies and other outcomes relating to COVID-19 [1, 2, 6]. Ethnicity intersects with occupation, with the social sorting of disadvantaged and minoritised ethnic groups into employment roles that have greater exposure to COVID-19 risks [3, 16]. For example, minoritized ethnic groups were less likely to have been furloughed since they were more likely to be found in essential occupations [15]. Future research should explore the intersecting pathways between occupation and ethnicity to improve our understanding of why these inequalities exist and which pathways we can tackle.

There are several limitations to our study. Not all participants had the same number of responses in the dataset (mean data points per individual of 6, standard deviation 2.2). While multi-level modelling is flexible and accommodates for this imbalance, we cannot rule out that it does not contribute bias to our data (e.g., if the number of responses was socially patterned). We do not account for all possible explanatory factors (e.g., deprivation, social distancing behaviours) that may explain occupational inequalities due to a lack of suitable data available for our analysis. Use of formal model building approaches (e.g., Directed Acyclic Graphs) or co-producing decisions with stakeholders could have improved this process. Through focusing on work sector, rather than specific occupation or role, we may be limited how generalisable our findings are. For example, teaching and education would include both primary and secondary teachers who were expected to teach classes face-to-face and therefore have different exposures to University lecturers who could far easier adapt to work from home. The lack of specific occupation categories may therefore under-estimate the specific risks and inequalities faced across England. Finally, our analyses are association-based and do not explore potential causal pathways or mechanisms through how and why occupation influences COVID-19 risk. Future research should extend our analyses to consider the specific mechanisms that may explain, mediate or moderate risk.

## Conclusions

Our study, using novel large-scale longitudinal data, demonstrates the importance of the social determinants of health through work and occupation in understanding the unequal burden in COVID-19 prevalence. We find complex and diverse pathways through which SARS-CoV-2 transmission may occur across numerous work sectors which can exacerbate, reinforce and create new health inequalities. Population groups employed in sectors with greater social contacts, less able to work from home or having front facing roles have greater likelihood of COVID-19. Additionally, groups that have experienced social and economic harms through furlough appear to have experienced a double jeopardy in greater likelihood of COVID-19. Evaluating whether these occupational-based inequalities have remained consistent or changed following the roll-out of vaccines, reorganisation of society 'back to normal', and continuing exposure to new variants/reinfections will be key for identifying how different occupational groups have experienced COVID-19.

## Supporting information

**S1 File. Appendix.**
(PDF)

## Acknowledgments

This work was produced using statistical data from ONS. The use of the ONS statistical data in this work does not imply the endorsement of the ONS in relation to the interpretation or analysis of the statistical data. This work uses research datasets which may not exactly reproduce National Statistics aggregates. We would also like to thank Francisco Rowe for their advice on our multi-level modelling methods.

## Author Contributions

**Conceptualization:** Mark A. Green, Malcolm G. Semple.

**Data curation:** Mark A. Green.

**Formal analysis:** Mark A. Green.

**Funding acquisition:** Mark A. Green.

**Investigation:** Mark A. Green, Malcolm G. Semple.

**Methodology:** Mark A. Green.

**Project administration:** Mark A. Green.

**Supervision:** Malcolm G. Semple.

**Validation:** Mark A. Green.

**Visualization:** Mark A. Green.

**Writing – original draft:** Mark A. Green.

**Writing – review & editing:** Mark A. Green, Malcolm G. Semple.

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
