## [Decision Letter · Decision Letter 0]

9 Jun 2022

PONE-D-21-18992

Occupational inequalities in the prevalence of COVID-19: A longitudinal observational study of England, August 2020 to January 2021

PLOS ONE

Dear Dr. Green,

Thank you for submitting your manuscript to PLOS ONE. After careful consideration, we feel that it has merit but does not fully meet PLOS ONE’s publication criteria as it currently stands. Therefore, we invite you to submit a revised version of the manuscript that addresses the points raised during the review process.

We are profoundly sorry for the time the review process has taken for this manuscript. We have noticed unnaturally low response rates to review invitations for COVID-19 and clinically related manuscripts recently. Please note that we have only been able to secure a single reviewer to assess your manuscript. We are issuing a decision on your manuscript at this point to prevent further delays in the evaluation of your manuscript. Please be aware that the editor who handles your revised manuscript might find it necessary to invite additional reviewers to assess this work once the revised manuscript is submitted. However, we will aim to proceed on the basis of this single review if possible.

The reviewer has raised a number of concerns that need attention. The reviewer has concerns over the study methodology and the use of surveillance data as primary data. One main concern the reviewer has is regarding the reporting in the study methodology, including the model development information. They also have major concerns regarding the conclusions and interpretations that can be drawn with selection bias in the study, and the outcome and significance that can be discussed in this context. The reviewer also has minor copyediting/structure concerns.

Could you please revise the manuscript to carefully address the concerns raised?

We look forward to receiving your revised manuscript.

Kind regards,

Sebastian Shepherd

Staff Editor

PLOS ONE

Journal Requirements:

2. Thank you for stating the following financial disclosure: "This work was supported by the Economic and Social Research Council [grant number ES/L011840/1]."

3. Thank you for stating the following in your Competing Interests section:  "None declared" 

Reviewers' comments:

Reviewer's Responses to Questions

**Comments to the Author**

1. Is the manuscript technically sound, and do the data support the conclusions?

Reviewer #1: Partly

2. Has the statistical analysis been performed appropriately and rigorously? 

Reviewer #1: No

3. Have the authors made all data underlying the findings in their manuscript fully available?

Reviewer #1: No

4. Is the manuscript presented in an intelligible fashion and written in standard English?

Reviewer #1: Yes

5. Review Comments to the Author

Reviewer #1: Overall results and discussion needs some more work. However, introduction and methodology need to be cleared.

The aim of this study is to evaluate how occupational inequalities in the prevalence of COVID-19 vary 59 across England and their possible explanatory factors, but there is almost nothing in the introduction section.

Moreover, how it is possible to develop interventions to minimize transmission risk for managing COVID-19 and 63 future pandemics by using the secondary data with limited variables and when selection bias is one of the biggest limitation.

I think it’s the data from the surveillance and not the primary data, as it was not collected for the research to develop interventions.

They need to change their outcome of this paper and its significance.

The model development information is missing; I could see univariate analysis only.

Please take of typo errors

In discussion authors never shared what intervention they are suggesting.

Discussion need to more structured and need to explain their results more clearly.

6. PLOS authors have the option to publish the peer review history of their article (what does this mean?). If published, this will include your full peer review and any attached files.

Reviewer #1: **Yes: **Tazeen Saeed Ali

---

## [Author Response · Author response to Decision Letter 0]

22 Jul 2022

Response to reviewers

Comment from authors: We would like to thank both the Editor and the Reviewer for their helpful comments and the opportunity to revise our paper further. We have worked hard on revising our paper in line with the suggestions which have strengthened the narrative and quality of the paper. If there are further changes please let us know. Below, we provide a point by point response to each comment (starting first with the Editorial comments, then the reviewer comments). Where page numbers are provided, we refer to the tracked changes version of the manuscript. Thank you again.

Journal Requirement Revisions:

Journal comment: Please ensure that your manuscript meets PLOS ONE's style requirements, including those for file naming. The PLOS ONE style templates can be found at 

Response: We have updated the manuscript to match the style requirements.

Journal comment: Thank you for stating the following financial disclosure: "This work was supported by the Economic and Social Research Council [grant number ES/L011840/1]."

Response: The funders had no role in study design, data collection and analysis, decision to publish, or preparation of the manuscript. We have edited the cover letter to make this clear now. We initially placed it in the acknowledgements but the PLOS One team suggested to not do this so we have removed it from there.

Journal comment: Thank you for stating the following in your Competing Interests section: "None declared" 

Response: The authors have declared that no competing interests exist. What have updated this on the system. We have updated this on the paper now.

Journal comment: In your Data Availability statement, you have not specified where the minimal data set underlying the results described in your manuscript can be found. PLOS defines a study's minimal data set as the underlying data used to reach the conclusions drawn in the manuscript and any additional data required to replicate the reported study findings in their entirety. All PLOS journals require that the minimal data set be made fully available. For more information about our data policy, please see http://journals.plos.org/plosone/s/data-availability.

Response: We have added more information so that data access is clear (this has been updated on the online system, as well as in the cover letter):

The ONS’s COVID-19 Infection Survey (CIS) is an individual level survey that collects population level data about people who live in the UK to estimate the prevalence of COVID-19. They collect a range of data including demographic and social data, as well as test everyone for COVID-19. These data contain sensitive and personal information about individuals (including information about date of birth and health information) . Such data cannot be openly shared to ensure privacy and meet statistical disclosure laws (i.e., researchers cannot legally distribute). 

Access requires approval by the ONS who act as the data owner and do not allow researchers to hold or distribute their data. Data are free to access for accredited researchers within the ONS’s Secure Research Service (SRS). The SRS is a trusted researcher environment which provides free access to the ONS’s sensitive data (it acts as a secure data repository). More information about the process, including how to apply to access the data, can be found here: https://www.ons.gov.uk/aboutus/whatwedo/statistics/requestingstatistics/secureresearchservice

All analytical code used to process the data and replicate the results in the paper can be found here: https://github.com/markagreen/occupational_inequalities_CIS

Metadata: https://ons.metadata.works/browser/dataset?id=293

Journal comment: We note that you have stated that you will provide repository information for your data at acceptance. Should your manuscript be accepted for publication, we will hold it until you provide the relevant accession numbers or DOIs necessary to access your data. If you wish to make changes to your Data Availability statement, please describe these changes in your cover letter and we will update your Data Availability statement to reflect the information you provide.

Response: This has been responded to and revised in the above/previous comment.

Journal comment: Please include captions for your Supporting Information files at the end of your manuscript, and update any in-text citations to match accordingly. Please see our Supporting Information guidelines for more information: http://journals.plos.org/plosone/s/supporting-information.

Response: These have now been added to the bottom of the manuscript (p26 tracked version) and we have updated citations within text accordingly.

Reviewer comments

Reviewer #1: Overall results and discussion needs some more work. However, introduction and methodology need to be cleared.

Reviewer comment: The aim of this study is to evaluate how occupational inequalities in the prevalence of COVID-19 vary across England and their possible explanatory factors, but there is almost nothing in the introduction section.

Response: We have now expanded the introduction to bring greater depth of the context of our study. We have emphasised making it clearer to the reader why this piece of work was important, what is novel and why it matters. We have also made reference to the importance of this work for informing intervention development, in line with other comments during your review. We have opted for a length similar to other PLoS One articles (i.e., try to be concise where possible). We have also changed the structure from three paragraphs, to 5 paragraphs so that it flows better and guides the reader through the importance of the problem we are investigating. This also helps accommodate the additional material. 

Specifically, additions include more detail on why it is important to look at occupational differences: 

“Most of this existing research has focused on severe outcomes or a narrow range of occupations/work sectors, meaning that we have less evidence of how SARS-CoV-2 infections vary across occupations or work settings. Preventing infections and exposure to SARS-CoV-2 will help to reduce severe COVID-19 outcomes. Understanding which occupations had higher or lower infections is imperative for designing preventative work-place interventions for managing COVID-19 and preparing for future pandemics.” (pp3-4 tracked version)

Additional text has been added on why occupation is important to consider if we want to understand inequalities:

“Occupational status intersects with age, sex, ethnicity and deprivation. For example, individuals from deprived neighbourhoods or Black and Asian ethnic groups are more likely to be employed in occupations that were unable to work from home or in ‘essential’ front facing roles (3,15); the same groups who have seen higher hospitalisations and mortality related to COVID-19. Occupation type is therefore a fundamental driver of exposure to SARS-CoV-2 in employed populations, meaning that occupation may partly explain or amplify health inequalities relating to COVID-19 (5,16).” (p4 tracked version)

We have also made some minor changes the wording of the introduction either to clarify statements (e.g., line 58 tracked version) or to state the importance for intervention development (lines 77-80 tracked version). 

We hope that this additional context, appended to the original introduction, brings it more in line with the expectations for an introduction.

Reviewer comment: Moreover, how it is possible to develop interventions to minimize transmission risk for managing COVID-19 and future pandemics by using the secondary data with limited variables and when selection bias is one of the biggest limitation. I think it’s the data from the surveillance and not the primary data, as it was not collected for the research to develop interventions.

Response:

The use of secondary data in epidemiology/public health is a common practice in studying population health issues (like our research question). It is a valid approach that has been used widely and PLoS One publishes a lot of papers that are based on secondary data analysis. Using secondary data here brings a lot of benefits to conducting our research project that would have not been possible if we had gone down a primary data route (especially in collecting a large data set as we needed). To help make this clearer in our paper, we have added a paragraph detailing this so that it is clear to the reader. The paragraph reads:

“The ONS Covid Infection Survey (CIS) was used as our data source. The CIS is a representative random sample survey of the population in England used to monitor trends in COVID-19 (18). While primarily used as a surveillance tool for COVID-19, CIS was also designed so that the data could be re-used by researchers as a secondary data resource for understanding population health issues relating to the pandemic. Secondary data are useful resources since they allow for efficient collection of large and complex data that saves time and costs over primary data collection (19). This is valuable for our study since (i) the rarity of our outcome (0.9% in this study) means we require a large sample size to find robust associations, and (ii) the CIS was designed to collect a national-level representative sample that minimises selection bias meaning our findings can be generalisable.” (p5 tracked version)

Selection bias, as described in the above paragraph, is not a big issue here since the secondary data source we used was a large national level survey that is representative and based on random sampling. This helps our results be generalisable and relevant; the representative design of the survey helps minimise issues of selection bias and is a strength of using the data source. 

We also note later that since all individuals are swabbed and tested for SARS-CoV-2, we minimise selection bias why relying on self-reported testing data. We have added a statement to make this clear now:

“This helps to minimise issues in secondary data based on self-reported testing records due to selection bias in who tests and who registers tests (18).” (lines 105-106 tracked version)

Secondary data has value to population health researchers beyond evaluating a single intervention. In our case, it allows us to understand the extent of occupational inequalities in COVID-19 risk and the reasons behind this. These issues are new and we lack detailed evidence on the reasons behind why certain occupational groups may be a higher/lower risk; important for learning from the pandemic. We can then use this information to try and design relevant interventions – our study is one piece of evidence in this process (this is no different to how social epidemiology and public health research operates too). We discuss specific additional changes and content about interventions later in the responses (see below/discussion section).

Reviewer comment: They need to change their outcome of this paper and its significance.

Response: The aim of our paper was “to evaluate how occupational inequalities in the prevalence of COVID-19 vary across England and their possible explanatory factors”. Our outcome variable is ‘whether an individual had a positive SARS-CoV-2 test or not’ which is the primary and established outcome measure for understanding prevalence of COVID-19 (i.e., a positive test is required to see who has COVID-19). We therefore believe that the correct outcome was used. 

Reviewer comment: The model development information is missing; I could see univariate analysis only.

Response: Our analyses included multivariable multi-level models (i.e., adjustment for all explanatory variables in the model). Please see Tables 1 and 2 which detail the results from the fully adjusted models here. Figures 6 and 7 were also generated from using fully adjusted models as well. To avoid confusion here to the reader, we have made it explicit in the statistical analysis section that we used multivariable models now (p7).

We have also added a further text to clarify our model development so that it is clear what we did. It now reads: “Multi-level binomial regression models were used to analyse the risk of COVID-19. Two models were used; one for employment status and one for occupational sector (our key exposure variables). Multivariable models were fully adjusted using a series of fixed effect variable representing our explanatory individual-level covariates that may explain differences in COVID-19 risk. Numeric values were z-score standardised to minimise issues with their different scales (age and household size). Two random effects are included: (i) participant ID (varying intercept) to account for repeat observations within the survey over time, and (ii) geographical area (varying intercept) to account for geographical inequalities in COVID-19 prevalence and therefore risk. Our model building process began through examining our key exposures and the random effects independently, prior to including all variables in the model, and then checking model assumptions. We present only the fully adjusted analyses here.” (p8 tracked version)

We have also mentioned why we calculated the conditional mean values from our multivariable models so it was clearer to the reader: “We estimate the conditional mean for each geographical area from our fully adjusted multivariable models. Through mapping these estimates, we can visually examine if there were distinct geographical patterns in COVID-19 during the study period to understand the role that location plays in our model and to identify the need for spatially targeted interventions.” (p8 tracked version)

Finally, we have moved the content about the interaction effect models for estimating changes over time that was originally present in the results into the statistical analysis section of the methodology. This improves the structure of the paper through having all the methodological steps in one place. We hope that all of these changes have help make things clearer now.

Reviewer comment: Please take of typo errors

Response: We have gone through the paper carefully and make changes to address spelling and grammar issues. We do not detail all of the changes here for brevity, but they can be found on the tracked changes version of the paper.

Reviewer comment: In discussion authors never shared what intervention they are suggesting.

Response:

Our study is a piece of descriptive epidemiology (observational study) rather than evaluating an intervention. We have a large range of results/key findings, meaning there are quite different ways to recommend interventions based on them. We agree with you that we need to make it clearer the implications of our work (including possible interventions). To make it clearer what interventions might be needed based on our study (since our findings speak to a range of issues and potential interventions arising from), we have expanded our discussion section to make this clearer to the reader. Specifically, when talking about furloughed populations we have added:

“Our findings would recommend careful educational messaging where future furlough schemes are introduced to help guide individuals who were furloughed in navigating their new risks, or encouraging employers to find ways to help individuals work from home rather than being furloughed where possible.” (p18 tracked version)

When talking about issues with student populations we have added:

“Improved ventilation and use of marks in indoor classroom settings during periods of high community prevalence may be needed to avoid similar patterns during term time (25). Such interventions may only be effective if paired with strategies around social experiences of University students.” (p19 tracked version)

We also note that we have a long discussion about the importance of workplace interventions. It would be beyond the scope of our article to provide bespoke interventions for all occupational groups, hence why we provide a more general discussion.

“Our findings suggest the need for better workplace interventions across diverse roles that can help contain COVID-19 transmission, whilst allowing individuals and employers to continue their social and economic activities (7). Occupational roles will need to further adapt to protect their employees from COVID-19. Minimising social contacts or mixing within occupational roles through sufficient preventative measures may be valuable. One study suggested that limiting the number of social contacts at work was the most important strategy for lowering the ‘R’ number if society keeps schools open (17). Repeat testing of employees may help to manage outbreaks, however testing behaviours can also widen inequalities (20). Support for lost earnings if individuals have to self-isolate will be key, especially as some of the work sectors identified here with higher prevalence (e.g., retail or hospitality) are characterised by low wages (16). The UK Government should also consider targeting particular work sectors considered at ‘higher risk’ of infections and introducing interventions in those areas when required. However, our findings of high prevalence of COVID-19 for furloughed and student populations demonstrates the need for broader strategies than just occupation-related interventions to help manage COVID-19 and tackle the drivers of health inequalities.” (pp19-20 on tracked version)

We have also stated in our key results paragraph that this work matters for intervention design: “with work status and occupational group being one route for interventions aimed at tackling inequalities” (p18 tracked version).

Our paper now hopefully makes it clear potential strategies or areas to improve in terms of interventions/policy development.

Reviewer comment: Discussion need to more structured and need to explain their results more clearly.

Response: We have gone through our discussion again and revised the content to try and improve the clarity of the text. Our structure follows the STROBE observational checklist – so first paragraph describe key results, then provide the interpretation for the key results (explain in detail each one separately), then describe limitations of your work, then conclusion.

---

## [Decision Letter · Decision Letter 1]

9 Jan 2023

PONE-D-21-18992R1Occupational inequalities in the prevalence of COVID-19: A longitudinal observational study of England, August 2020 to January 2021PLOS ONE

Dear Dr. Green,

Thank you for submitting your manuscript to PLOS ONE. After careful consideration, we feel that it has merit but does not fully meet PLOS ONE’s publication criteria as it currently stands. Therefore, we invite you to submit a revised version of the manuscript that addresses the points raised during the review process. Your paper has been re-reviewed. Two of the reviewers have provided some further requests to the authors, especially in terms of context given to the readers, interpretation, and confounding factors potentially affecting the study. You will find all these comments below.However, I would like to point out that I am still concerned because one of the reviewers assessing your paper has suggested its rejection on the basis of serious statistical issues (please see comments provided by Reviewer # 3). Given that I have split opinions on your manuscript, but that the improvements previously done are mostly satisfying, I would like to give you another opportunity to respond to the reviewers. Although it is evident that a major effort must be done in regard to Reviewer # 3, please note that all the comments and suggestions provided in these review reports require your attention. Therefore, please respond to all of them, offering suitable and well-supported rationales and amendments.

We look forward to receiving your revised manuscript.

Kind regards,

Sergio A. Useche, Ph.D.

Academic Editor

PLOS ONE

Reviewers' comments:

Reviewer's Responses to Questions

**Comments to the Author**

1. If the authors have adequately addressed your comments raised in a previous round of review and you feel that this manuscript is now acceptable for publication, you may indicate that here to bypass the “Comments to the Author” section, enter your conflict of interest statement in the “Confidential to Editor” section, and submit your "Accept" recommendation.

Reviewer #2: (No Response)

Reviewer #3: (No Response)

Reviewer #4: (No Response)

2. Is the manuscript technically sound, and do the data support the conclusions?

Reviewer #2: Yes

Reviewer #3: No

Reviewer #4: Partly

3. Has the statistical analysis been performed appropriately and rigorously? 

Reviewer #2: Yes

Reviewer #3: No

Reviewer #4: Yes

4. Have the authors made all data underlying the findings in their manuscript fully available?

Reviewer #2: Yes

Reviewer #3: No

Reviewer #4: No

5. Is the manuscript presented in an intelligible fashion and written in standard English?

Reviewer #2: Yes

Reviewer #3: Yes

Reviewer #4: Yes

6. Review Comments to the Author

Reviewer #2: The paper is interesting and clear. As a minor note, please add explanation to ONS (what does it stand for?). Please also check supplementary files for the definition of "hospitality sector" - there seems to be an error (examples of military occupations). Moreover, in supplementary map figures (Fig C), please clarify the title (the conditional mean of what?).

Reviewer #3: The manuscript is clearly written and the authors appear to have responded well to the previous reviewer’s concern. Nevertheless, I believe that the paper has a serious statistical flaw. Respondents in the sample have contributed between one and 13 observations to the data file, depending on the number of times that they performed Covid tests. That is, different respondents contribute different numbers of observations to the sample and hence some respondents contribute far more weight than others to the estimates. The authors include random effects in their models, but this does not take care of the issue (it does account for clustering and would be fine if all respondents had equal numbers of observations in the sample). This problem affects all estimates – simple descriptive statistics and model estimates. For example, it could be the case that the estimates of positivity rates are higher for younger people (or for those furloughed etc.) primarily because they are more likely than others to provide a large number of tests.

It is also unclear how selection bias is ‘minimized.’ Selection bias is not reduced by having a very large sample. What is important is having a random sample along with low non-response. The authors have not provided any non-response rate or indicated whether non-response varies across characteristics of interest. It is possible, for example, that persons with the most exposure to infection were the most likely to agree to take part in the survey.

Reviewer #4: I am enthusiastic about this study. In particular, I am very impressed with the data source and the thoroughness of the analysis. I congratulate the authors on a fascinating study, and apologize for any misunderstandings on my part in the questions I raise below.

MAJOR COMMENTS

1. The authors include a lot of interesting information in this paper, and paradoxically, I think that doing so potentially opens the paper up to critique. Speaking for myself, I must admit that, as interesting as the findings were, I was left wondering about the "so what." The authors state that their aim is to understand how occupational inequities varied across England, including across time. But why? The time trends by work/occupation would appear to mirror overall trends, for example. Do the authors envision interventions that are both work and time-specific? (And if so, I'm not sure if I'm fully convinced on the feasibility!) When I think about occupation and time, I think first of whether inequalities diminished in the vaccine era (not possible to examine in this study due to the time window) or the era of variants.

2. Related to the above: I must admit I was left wondering what the purpose of the geographical analysis is. Notably, the authors do not show occupational inequality by geography. So, the analysis seems to actually be outside of the stated aim. But, even if the authors did show occupational inequality by geography, what is the implication? It seems that a more nuanced analysis would be required to understand why those variations occur. Differences in occupations held, for example. To be clear: I think the analysis and visualizations are neat. But, it's not clear to me that they fit within this paper.

3. "We need to better understand how risk varies across these groups to better develop interventions," in Introduction: Also related to the above, I must admit I was not very convinced by this argument. It's hard for me to think of age-specific interventions aside from vaccination, and in fact the authors themselves mention (prior paragraph) an interest in non-vaccination interventions.

4. Methods: I was unable to find information on the timing of the exposure of interest relative to the outcome of interest. Do the authors know for certainty if the exposure necessarily predated infection? For example, is it conceivable that workers were furloughed after, rather than before, becoming infected? Can the authors clarify the recall period for the exposure variable?

5. Figures: It's not necessarily clear whether other factors are adjusted for in obtaining these estimates. I suggest clarifying this in the figure titles/captions.

6. differences by sex in analysis of work sector, in Results: It is unclear to me what these conclusions are based on. Direct statistical comparison of the sexes is in theory possible for each sector, but such tests are not reported here.

7. several places, in Results: "limited conclusions can be drawn since confidence intervals overlapped." The authors could in theory test these comparisons (see above), so I was surprised by this caveat. I suggest that the authors either (1) perform the formal statistical comparison or (2) simply omit the sentence about the limitation (fully avoiding the topic).

8. Methods: "model building process": Can the authors elaborate on what the process was? In other words, how did the authors decide on the variables to include? I realize that some of this is provided in the variable list, so maybe this is partly about slightly rearranging the text. More broadly, I'm curious if the authors employed a DAG-like approach in selecting the variables. Some specific variables included/excluded might raise concerns/questions, especially in the absence of such clarity.

9. It might be helpful providing some context, whether using descriptives from the data or citation of other data, on who is being furloughed. I'm curious in particular what sorts of occupations people tend to be furloughed from. Also, is there any information on activities of individuals during furlough time? Are they taking on side jobs, for example?

MINOR COMMENTS

10. Abstract: The authors report here that a "large proportion of SARS-CoV-2 transmission occurs outside of work." This is not necessarily consistent with the sentence that follows, "populations who experienced social and economic harms through being furloughed were also more likely to experience." Note that the first sentence is about P(furlough|infection) while the concluding sentence is about P(infection|furlough). Given that the authors found a higher prevalence among furloughed workers (ie a high P(infection|furlough)), it seems that an easy fix would be to simply modify the first sentence to reflect that finding.

11. "In comparison to individuals who were employed, individuals who were furloughed had 81% higher odds," Results: Given that the employment-status measurement is presumably time dependent, the comparison here is technically not comparing groups of individuals but groups of person-time, yes? I have the same question for other areas of the text where similar language is used but the exposure is time dependent (including the Results text for Table 2, I think).

12. Methods, z-score standardization: It's not clear to me what "issues" the authors are referring to here. My understanding has been that standardization is generally mostly done for interpretation. So, the decision of standardization strikes me as a little surprising since the authors standardized non-exposure covariates (and we might not be particularly interested in reporting such effects). Can the authors clarify? I'm happy to acknowledge there's something I'm missing here, or some difference in perspective. Alternatively, I suggest that the authors simply omit the statement.

13. Figure by age: The authors might consider grouping the ages. Even with 5-year age intervals, the bounds should be narrower but the age bins should be small enough to understand the shape of the curve.

14. "adequate strategies", Discussion: I suggest including at least some examples of adequate strategies here in parentheses or at least including some citations that point to such strategies.

15. discussion on ethnicity and attenuation: I think this section of the discussion may raise concerns from some readers since the attenuation the authors refer to doesn't derive from a model that was specifically designed to examine this. Moreover, some researchers critique the Baron-and-Kenny method for examining mediation.

7. PLOS authors have the option to publish the peer review history of their article (what does this mean?). If published, this will include your full peer review and any attached files.

Reviewer #2: No

Reviewer #3: No

Reviewer #4: No

---

## [Author Response · Author response to Decision Letter 1]

14 Feb 2023

Response to reviews

Editor

To the editor, thank you for the opportunity to revise our paper. In sum, the comments are largely positive and we have responded to them individually which has improved the quality of our paper. We provide a point-by-point rebuttal below (where we refer to lines to show revisions, we refer to the tracked changes version of the paper). We note here that you highlighted the statistical issues of our paper. Reviewer 2 and 4 suggest that the statistical analysis has been performed appropriately and rigorously. Reviewer 3 offers more critical comments, however we believe that they are incorrect and have provided a rebuttal to their comments that shows why our results are robust. If you require any further information, please do not hesitate to get in contact.

Reviewer #2

Comment: The paper is interesting and clear. 

Response: We would like to thank you for your kind and positive comments about our paper.

Comment: As a minor note, please add explanation to ONS (what does it stand for?). 

Response: We apologise not for explaining the acronym. We have defined this now in the abstract (see line 21 tracked changes version) and the first occurrence in the main text of the paper (see line 91 tracked changes version).

Comment: Please also check supplementary files for the definition of "hospitality sector" - there seems to be an error (examples of military occupations). 

Response: Apologies for this mistake. This was a copy and paste error. We have now updated the description to read: “Server, waiting staff, bar staff, chef” (see p1 Appendix).

Comment: Moreover, in supplementary map figures (Fig C), please clarify the title (the conditional mean of what?).

Response: We have removed this figure now from the Appendix, based on the suggestion from Reviewer #4 to leave out the geographical maps from the article.

Reviewer #3

Comment: The manuscript is clearly written and the authors appear to have responded well to the previous reviewer’s concern. 

Response: We thank you for the kind comments about the quality of our paper and revisions here. 

Comment: Nevertheless, I believe that the paper has a serious statistical flaw. Respondents in the sample have contributed between one and 13 observations to the data file, depending on the number of times that they performed Covid tests. That is, different respondents contribute different numbers of observations to the sample and hence some respondents contribute far more weight than others to the estimates. The authors include random effects in their models, but this does not take care of the issue (it does account for clustering and would be fine if all respondents had equal numbers of observations in the sample). This problem affects all estimates – simple descriptive statistics and model estimates. For example, it could be the case that the estimates of positivity rates are higher for younger people (or for those furloughed etc.) primarily because they are more likely than others to provide a large number of tests.

Response: We would like to start by saying that we would like our response to be in the spirit of being constructive and not be interpreted as being negative, since we appreciate that as a reviewer you can coming from a point of view that is to help us the authors in improving the quality of our paper. 

The description that this is a serious statistical flaw here is not true or accurate. 

A key advantage of multi-level modelling is to appropriately handle imbalance in the number of observations per group (i.e., records per individual in our paper). Multi-level modelling was designed to leverage existing information to generate estimates for data observations which may not be captured in the sample. The concept of “random effects” reflects this, i.e., that some observations may not be observed but estimates can be generated drawing on existing data points and accepting that there is uncertainty or randomness in the estimates (based on relatively forgiving missing at random assumptions). In that sense, the technique is designed to minimise the issues you describe. 

Simulation studies investigating the issue you described have showed how robust multi-level modelling? is to any imbalance in groups sample sizes. For example, Cools et al. (2009) say that “imbalance could be ignored” with simulations showing a greater amount of imbalance than that in our study still resulted in robust results. Indeed even more complex versions of multi-level models than ours have shown that unbalance doesn’t really have a huge influence (e.g., see Milliren et al. 2018). The standard errors in models tend to be larger because of missing data, which means our associations are probably conservative as a result.

We noted in the ‘Data’ subsection that the range of number of responses per individual was 1-13. Mean number of responses was 6 with a standard deviation of 2.2. So the majority of data is clustered around the mean quite well (61% of people have monthly data so 6 observations). Data collection periods were driven by contact from the ONS (i.e., people could not choose to send in more tests). This distribution would help the robustness of any imbalance. A mean of 6 is the often cited required level for generated unbiased estimates of random effects variance parameters (not our focus in the results but worth mentioning). Our large sample size for individuals helps to minimise any bias in the variation of observations per individual as well.

It is also worth noting that many published multi-level applications have similar imbalance issues as we describe, including those published in PLoS One, which are robust because of the way the method can handle data imbalances.

We do accept that there may be bias in our dataset relating to what you have mentioned (i.e., socially patterned imbalance in number of responses), but the modelling framework we deploy handles any imbalances. We therefore feel that in the spirit of peer review, your comment has value and is important to note (even if our choice of statistical model handles the issue well). We have made a note of this in the limitations section of the paper now to accommodate this as a potential source of bias in our study: “Not all participants had the same number of responses in the dataset (mean data points per individual of 6, standard deviation 2.2). While multi-level modelling is flexible and accommodates for this imbalance, we cannot rule out that it does not contribute bias to our data (e.g., if the number of responses was socially patterned).” (lines 403-406 tracked changes version)

We recognise that data imbalance might affect descriptive statistics. We investigated this by restricting observations to a single observation per person. We ran into some difficulty here as we would have to define where the single observation from (e.g., taking first observation means most have data taken from the start of the period with low population incidence of cases resulting in misleading descriptives). In sum, we tried many different definitions (e.g., random selection, selecting once in a month and repeating the descriptive statistics for each month) but it did not materially change any of our findings or conclusions. We have therefore left the descriptives as they were since it does not have an impact. 

We hope that this is a sufficient response and rebuttal. We do not feel that a full description of this discussion is required in the paper, but hope that you can be reassured that the results are robust and that we have acknowledge the issue you raised.

References

Cools et al. 2009. Design efficiency for imbalanced multilevel data. Behavior Research Methods, 41(1), 192-203. https://link.springer.com/article/10.3758/BRM.41.1.192

Milliren et al. 2018. Does an uneven sample size distribution across settings matter in cross-classified multilevel modeling? Results of a simulation study. Health and Place 52: 121-126, https://www.ncbi.nlm.nih.gov/pmc/articles/PMC6171360/

Comment: It is also unclear how selection bias is ‘minimized.’ Selection bias is not reduced by having a very large sample. What is important is having a random sample along with low non-response. The authors have not provided any non-response rate or indicated whether non-response varies across characteristics of interest. It is possible, for example, that persons with the most exposure to infection were the most likely to agree to take part in the survey.

Response: We did not state in our paper that a large sample minimised selection bias. We only talk about sample size when mentioning about the rarity of our outcome (i.e., the need to have a large enough sample to capture enough events). We did mention that the survey we used was designed to “collect a national-level representative sample that minimises selection bias meaning our findings can be generalisable” (lines 99-100 tracked changes version). This is an important point to raise here in response to your comment. The ONS worked hard to ensure a generalisable and representative sample – important in helping our results be meaningful. The survey is a random sample survey as well, which you describe above as important. We think that it meets you requirement.

The dataset we used, the ONS Covid Infection Survey, were provided as secondary data. We added in a statement in a previous review about the value of using secondary data (see lines 93-95 tracked changes version). We did not have access to sampling frame, choices or attrition information as we were not running the study. In December 2020, within our study period, the ONS reported that attrition was 0.62% - very low (see https://www.ons.gov.uk/peoplepopulationandcommunity/healthandsocialcare/conditionsanddiseases/methodologies/covid19infectionsurveypilotmethodsandfurtherinformation or reference 18 in the paper). We have now added a statement into the paper about low attrition that we hope can accommodate your broader point: “Attrition from the survey was low, reported as 0.62% in December 2020, supported by monthly data refreshes of new participants in response to the attrition and ensure the representativeness of the sample (18).” (lines 116-119 tracked changes version)

No data are perfect, however we hope that you would agree that the dataset we used for our analysis offers value for answering our research questions.

Reviewer #4

Comment: I am enthusiastic about this study. In particular, I am very impressed with the data source and the thoroughness of the analysis. I congratulate the authors on a fascinating study, and apologize for any misunderstandings on my part in the questions I raise below.

Response: Thank you for your kind and supportive comments about the quality of our paper. We also appreciate the tone and writing style of your review, which we found to be positive, honest and constructive. 

MAJOR COMMENTS

Comment: 1. The authors include a lot of interesting information in this paper, and paradoxically, I think that doing so potentially opens the paper up to critique. Speaking for myself, I must admit that, as interesting as the findings were, I was left wondering about the "so what." The authors state that their aim is to understand how occupational inequities varied across England, including across time. But why? The time trends by work/occupation would appear to mirror overall trends, for example. Do the authors envision interventions that are both work and time-specific? (And if so, I'm not sure if I'm fully convinced on the feasibility!) When I think about occupation and time, I think first of whether inequalities diminished in the vaccine era (not possible to examine in this study due to the time window) or the era of variants.

Response: This is an interesting and thoughtful comment. We have revised our paper throughout to help tighten our narrative in line with this suggestion. We tackle this in three main sections described below.

First, we note that our introduction section does not mention the ‘time’ element to our study nor is mentioned in our aim. As such, it is important that we make sure the rest of our paper is consistent with this. We have added an additional statement to clarify that we do look at time, to signpost the rest of our analyses. We added: “Finally, we consider how trends in occupational inequalities changed throughout our study period to examine if certain time periods produced differences in infection risk across occupational groups.” (lines 85-87 tracked changes version) We further add the following statement to the ‘statistical analyses’ subsection later too which helps to clarify why we looked at time: “This was done to examine if any occupational group experienced differences in infections at certain time periods which might have not been consistent across work sectors (e.g., differences in restrictions or lockdowns affecting who could work from home).” (lines 194-196 tracked changes version) Presenting the data by time period is still valuable, even if (as you rightly say) we need to play down its meaning in the paper. We feel that this additional sentence helps make the narrative consistent, especially when combined with the additional statements below that help to downplay some of the narrative. 

Second, we do give two examples in the discussion where time and economic activity come together through (i) students at the start of term (lines 353-360 tracked changes version) and (ii) early spikes in people employed in personal services (lines 368-371 tracked changes version). However, we accept your caution in making too much about this. We have added a statement into the paper to try and minimise any incorrect conclusions being drawn here: “We also suggest caution in any interpretation that there might be different associations between occupational categories over time. Most measures of work/occupation we analysed closely followed overall population-level trends (i.e., there was no sequencing of transmission between occupational groups). Time-specific interventions targeted at particular occupational groups may therefore be ineffective or unfeasible.” (lines 371-375 tracked changes version)

Third, we really liked your suggestion over future research, We can explore the exact questions you identify with newer versions of these data. We think it is a fascinating idea to see how things have changed over time with the roll out of vaccines or the era of new variants. We thank you for encouraging us here. With this in mind, we have added the following statements to the conclusion which we hope help to highlight the topics you suggest and encourage others to research. Specifically, we have added: “Evaluating whether these occupational-based inequalities have remained consistent or changed following the roll-out of vaccines, reorganisation of society ‘back to normal’, and continuing exposure to new variants/reinfections will be key for identifying how different occupational groups have experienced COVID-19.” (lines 430-434 tacked changes)

Comment: 2. Related to the above: I must admit I was left wondering what the purpose of the geographical analysis is. Notably, the authors do not show occupational inequality by geography. So, the analysis seems to actually be outside of the stated aim. But, even if the authors did show occupational inequality by geography, what is the implication? It seems that a more nuanced analysis would be required to understand why those variations occur. Differences in occupations held, for example. To be clear: I think the analysis and visualizations are neat. But, it's not clear to me that they fit within this paper.

Response: Thank you for your feedback. Upon reflection following your comment, we agree that the geographical analysis does not really fit in with the narrative of paper. We have removed it from the paper now. To remedy this, we have made the following changes.

First, we have revised our ‘Materials and Methods’ section. In the ‘Data’ subsection, we have removed ‘geographical location’ from the list of explanatory variables (lines 158-161). In the ‘Statistical analyses’ we revised the description of the random effect for geographical location. This was necessary to revise its description to be clearer, as well as to add some of the material we deleted from the ‘data’ subsection which still required reporting. It now reads: “(ii) geographical area of residence (varying intercept). While we do not report results by geographical location here, it was included to account for the spatial heterogeneity in COVID-19 outcomes observed in England (2). Regions (n=116) were created by the ONS and match Local Authority districts, with districts combined to make sure no region has a population of less than 500,000 to preserve data security.” (lines 174-180 tracked changes version) We also removed the entire description of how the maps were produced from the models (see lines 184-191 tracked changes version)

Second, we have removed all mention of the geographical data from the ‘Results’ section (see lines 288-295 tracked changes version).

Third, we have removed all mention of these findings from the ‘Discussion’ section (see lines 333-335 tracked changes version). 

Fourth, we have removed the maps from the Appendix.

We note here that there was no real mention of this analysis, or justification why it was needed, in the introduction. As such, the changes have made our paper a bit more tighter and focused. Thank you for your kind comments about our plots though – we liked them, even if they do not ultimately fit in here.

Comment: 3. "We need to better understand how risk varies across these groups to better develop interventions," in Introduction: Also related to the above, I must admit I was not very convinced by this argument. It's hard for me to think of age-specific interventions aside from vaccination, and in fact the authors themselves mention (prior paragraph) an interest in non-vaccination interventions.

Response: We have removed this statement in line with your suggestion. We have also revised the previous statement to remove the age comment and make it more general. It now reads: “First, there is a paucity of evidence on the extent of risks of COVID-19 by granular occupational groups or work sectors.” (lines 78-79 tracked changes version)

Comment: 4. Methods: I was unable to find information on the timing of the exposure of interest relative to the outcome of interest. Do the authors know for certainty if the exposure necessarily predated infection? For example, is it conceivable that workers were furloughed after, rather than before, becoming infected? Can the authors clarify the recall period for the exposure variable?

Response: The survey (exposure and controls) and swab (outcome) data are collected at each time point. So they represent the current employment status at data collection at the time the nose swab is taken which is then tested for SARS-CoV-2. This means we have good correspondence between exposure and outcome at the time of data collection. In some rare instances, people might have changed job in the last week prior to the survey where they were infected, but this is highly unlikely. It also means that as people’s status changes, records are updated over time. We have added an additional sentence into the ‘Data’ subsection to make this clear: “The survey and SARS-CoV-2 tests were completed together at each data collection time point.” (lines 105-106 tracked changes version)

Comment: 5. Figures: It's not necessarily clear whether other factors are adjusted for in obtaining these estimates. I suggest clarifying this in the figure titles/captions.

Response: Figures 1-5 are all unadjusted. We have now added ‘unadjusted’ into each title caption so this is clear. Figures 6 and 7 were both adjusted and already contained this in the title captions.

Comment: 6. differences by sex in analysis of work sector, in Results: It is unclear to me what these conclusions are based on. Direct statistical comparison of the sexes is in theory possible for each sector, but such tests are not reported here.

Response: We have removed this statement as it was not formally tested (see lines 241-246).

Comment: 7. several places, in Results: "limited conclusions can be drawn since confidence intervals overlapped." The authors could in theory test these comparisons (see above), so I was surprised by this caveat. I suggest that the authors either (1) perform the formal statistical comparison or (2) simply omit the sentence about the limitation (fully avoiding the topic).

Response: We have taken your advice and removed all of these sentences (see lines 218-219, 244-246, 306-307 tracked changes version). 

Comment: 8. Methods: "model building process": Can the authors elaborate on what the process was? In other words, how did the authors decide on the variables to include? I realize that some of this is provided in the variable list, so maybe this is partly about slightly rearranging the text. More broadly, I'm curious if the authors employed a DAG-like approach in selecting the variables. Some specific variables included/excluded might raise concerns/questions, especially in the absence of such clarity.

Response: There was no formal model building process. It was more informed from discussions with external stakeholders as part of the project, as well as informed by the literature and what was available in the survey. Since we did use a formal process like a DAG, we have deleted this text to avoid any confusion here (see lines 180-182 tracked changes version). 

Additionally, we have added this as a limitation to the paper when talking about choice of variables. We added the following statement: “Use of formal model building approaches (e.g., Directed Acyclic Graphs) or co-producing decisions with stakeholders could have improved this process.” (lines 408-410 tracked changes version)

Comment: 9. It might be helpful providing some context, whether using descriptives from the data or citation of other data, on who is being furloughed. I'm curious in particular what sorts of occupations people tend to be furloughed from. Also, is there any information on activities of individuals during furlough time? Are they taking on side jobs, for example?

Response: We think that any additional analysis here is beyond the aim of paper (especially as we already have quite a lot of results to cover). We have therefore opted to add in some additional details and references into the discussion to help the interpretation of these findings. Specifically, we have added: “Evidence has showed that people who were low-income backgrounds were more likely to have been furloughed (15).” (lines 343-344 tracked changes version) We extended on this point later in the discussion when talking about ethnic inequalities as well: “For example, minoritized ethnic groups were less likely to have been furloughed since they were more likely to be found in essential occupations (15).” (lines 399-400 tracked changes version) 

We could not find any studies or evidence on whether people who were furloughed got side jobs or other specific activities during their furlough time – this is a gap in the literature. We do note in our discussion that social contacts were higher than in non-furloughed groups: “ While furloughed populations may have fewer work social contacts, evidence suggests that their leisure and social contacts were higher than other groups (23).” (lines 342-343 tracked changes version)

We also added to the discussion more recent evidence showing mental health decline in people who were furloughed. We added: “Evidence elsewhere has suggested that people who were furloughed also experienced a small decline in their mental health (24).” (lines 348-349 tracked changes version)

MINOR COMMENTS

Comment: 10. Abstract: The authors report here that a "large proportion of SARS-CoV-2 transmission occurs outside of work." This is not necessarily consistent with the sentence that follows, "populations who experienced social and economic harms through being furloughed were also more likely to experience." Note that the first sentence is about P(furlough|infection) while the concluding sentence is about P(infection|furlough). Given that the authors found a higher prevalence among furloughed workers (ie a high P(infection|furlough)), it seems that an easy fix would be to simply modify the first sentence to reflect that finding.

Response: We agree with the comment and have revised the statement as you suggest. It now reads: “While our findings demonstrate the need for greater workplace interventions to protect employees tailored to their specific work sector needs, focusing on employment alone ignores the importance of SARS-CoV-2 transmission outside of employed work (i.e., furloughed and student populations).” (lines 31-36 tracked changes version)

Comment: 11. "In comparison to individuals who were employed, individuals who were furloughed had 81% higher odds," Results: Given that the employment-status measurement is presumably time dependent, the comparison here is technically not comparing groups of individuals but groups of person-time, yes? I have the same question for other areas of the text where similar language is used but the exposure is time dependent (including the Results text for Table 2, I think).

Response: Yes correct, the employment measure will be time dependent. Writing the language clearly is tricky, so we have opted to simply add in a statement to make this clear to the reader. It reads: “Here we describe groups of person-time as the measure is time dependent.” (lines 255 and 273 tracked changes version)

Comment: 12. Methods, z-score standardization: It's not clear to me what "issues" the authors are referring to here. My understanding has been that standardization is generally mostly done for interpretation. So, the decision of standardization strikes me as a little surprising since the authors standardized non-exposure covariates (and we might not be particularly interested in reporting such effects). Can the authors clarify? I'm happy to acknowledge there's something I'm missing here, or some difference in perspective. Alternatively, I suggest that the authors simply omit the statement.

Response: We were concerned that some variables were measured on different scales and it may be that the different scales/ranges of values may introduce some bias into our models (e.g., wider or smaller ranges having stronger associations that are merely a product of their scale). We acknowledge that there are differences in opinion here, with no consensus on correct practice which makes it a more subjective choice. As such, we have taken your suggestion to simply omit the sentence for ease and to avoid any further confusion through mention of ‘issues’. The sentence now reads: “Numeric values were z-score standardised (age and household size).” (line 172)

Comment: 13. Figure by age: The authors might consider grouping the ages. Even with 5-year age intervals, the bounds should be narrower but the age bins should be small enough to understand the shape of the curve.

Response: We have revised Figure 2 based on your feedback. We now use 5-year age bands (bar 18-24 which we combined into a single rather than having 18-19, 20-24). We also made some minor edits to the text in the results to match the updated plot (e.g., 18-22 changed to 18-24).

Comment: 14. "adequate strategies", Discussion: I suggest including at least some examples of adequate strategies here in parentheses or at least including some citations that point to such strategies.

Response: The purpose of this paragraph was to summarise our key findings of the paper at the start of the discussion. We have decided that following your comment, it was better to remove the “for adequate strategies” since it was vague and less tied to our findings (see line 337 tracked changes version). Rather, we feel having potential strategies or interventions mention alongside our interpretation of relationships subsequently in the discussion, rather than a general throwaway statement at the start. We hope this helps meet your suggestion.

Comment: 15. discussion on ethnicity and attenuation: I think this section of the discussion may raise concerns from some readers since the attenuation the authors refer to doesn't derive from a model that was specifically designed to examine this. Moreover, some researchers critique the Baron-and-Kenny method for examining mediation.

Response: We have removed any suggestion of attenuation following your comment which we agree with (see lines 394-397 in tracked changes version). This means the paragraph is a bit sharper in focus to describing ethnic inequalities.

---

## [Decision Letter · Decision Letter 2]

3 Mar 2023

Occupational inequalities in the prevalence of COVID-19: A longitudinal observational study of England, August 2020 to January 2021

PONE-D-21-18992R2

Dear Dr. Green,

We’re pleased to inform you that your manuscript has been judged scientifically suitable for publication and will be formally accepted for publication once it meets all outstanding technical requirements.

Kind regards,

Sergio A. Useche, Ph.D.

Academic Editor

PLOS ONE

Additional Editor Comments (optional):

Thanks for the amendments made. The paper can be considered for acceptance in its current form.

Reviewers' comments:

Reviewer's Responses to Questions

**Comments to the Author**

1. If the authors have adequately addressed your comments raised in a previous round of review and you feel that this manuscript is now acceptable for publication, you may indicate that here to bypass the “Comments to the Author” section, enter your conflict of interest statement in the “Confidential to Editor” section, and submit your "Accept" recommendation.

Reviewer #2: All comments have been addressed

Reviewer #3: (No Response)

2. Is the manuscript technically sound, and do the data support the conclusions?

Reviewer #2: Yes

Reviewer #3: Yes

3. Has the statistical analysis been performed appropriately and rigorously? 

Reviewer #2: Yes

Reviewer #3: Yes

4. Have the authors made all data underlying the findings in their manuscript fully available?

Reviewer #2: Yes

Reviewer #3: Yes

5. Is the manuscript presented in an intelligible fashion and written in standard English?

Reviewer #2: Yes

Reviewer #3: Yes

6. Review Comments to the Author

Reviewer #2: The authors have addressed all my comments. XXXXXXXXXXXXXXXXXXXXXXXXXXXXXXXXXXXXXXXXXXXXXXXXXXXXXXXXX

Reviewer #3: The authors have been very responsive to extensive reviewers’ comments through two sets of revisions and the paper is stronger as a result. The paper will make a useful contribution to the Covid literature.

7. PLOS authors have the option to publish the peer review history of their article (what does this mean?). If published, this will include your full peer review and any attached files.

Reviewer #2: No

Reviewer #3: No

---

## [Editor Report · Acceptance letter]

8 Mar 2023

PONE-D-21-18992R2 

Occupational inequalities in the prevalence of COVID-19: A longitudinal observational study of England, August 2020 to January 2021 

Dear Dr. Green:

I'm pleased to inform you that your manuscript has been deemed suitable for publication in PLOS ONE. Congratulations! Your manuscript is now with our production department. 

Kind regards, 

on behalf of

Dr. Sergio A. Useche 

Academic Editor

PLOS ONE